# Human DICER helicase domain recruits PKR and modulates its antiviral activity

**Thomas C. Montavon**[1]☯, **Morgane Baldaccini**[1]☯, **Mathieu Lefèvre**[1]☯, **Erika Girardi**[1], **Béatrice Chane-Woon-Ming**[1], **Mélanie Messmer**[1], **Philippe Hammann**[2], **Johana Chicher**[2], **Sébastien Pfeffer**[1]*

**1** Université de Strasbourg, CNRS, Architecture et Réactivité de l'ARN, UPR9002, Strasbourg, France,
**2** Université de Strasbourg, CNRS, Institut de Biologie Moléculaire et Cellulaire, Plateforme Protéomique Strasbourg–Esplanade, Strasbourg, France

☯ These authors contributed equally to this work.
* s.pfeffer@ibmc-cnrs.unistra.fr

## Abstract

The antiviral innate immune response mainly involves type I interferon (IFN) in mammalian cells. The contribution of the RNA silencing machinery remains to be established, but several recent studies indicate that the ribonuclease DICER can generate viral siRNAs in specific conditions. It has also been proposed that type I IFN and RNA silencing could be mutually exclusive antiviral responses. In order to decipher the implication of DICER during infection of human cells with alphaviruses such as the Sindbis virus and Semliki forest virus, we determined its interactome by proteomics analysis. We show that DICER specifically interacts with several double-stranded RNA binding proteins and RNA helicases during viral infection. In particular, proteins such as DHX9, ADAR-1 and the protein kinase RNA-activated (PKR) are enriched with DICER in virus-infected cells. We demonstrate that the helicase domain of DICER is essential for this interaction and that its deletion confers antiviral properties to this protein in an RNAi-independent, PKR-dependent, manner.

## Author summary

While RNAi has been recognized as an efficient antiviral defense system in organisms such as plants and insects, its physiological importance in mammals remains to be determined. DICER is an enzyme involved in cleaving long double-stranded RNAs and is essential for RNAi induction. Using mass spectrometry analysis, we determined its interactome in human cells and showed that RNA binding proteins such as PKR are specifically enriched upon infection with the Sindbis virus or the Semliki forest virus. We determined that the N terminal helicase domain of the DICER protein acts as a platform to recruit these factors during infection and that its deletion confers an antiviral activity to DICER.

**Data Availability Statement:** The mass spectrometry proteomics data have been deposited to the ProteomeXchange Consortium via the

PRIDE partner repository with the dataset identifier PXD019093 and 10.6019/PXD019093.

**Funding:** This work was funded by the European Research Council (ERC-CoG-647455 RegulRNA) (to SP) and was performed under the framework of the LABEX: ANR-10-LABX-0036_NETRNA (to SP) and ANR-17-EURE-0023 (to SP), which benefits from a funding from the state managed by the French National Research Agency as part of the Investments for the future program. This work has also received funding from the People Programme (Marie Curie Actions) of the European Union's Seventh Framework Program (FP7/2007-2013) under REA grant agreement n° PCOFUND-GA-2013-609102, through the PRESTIGE program coordinated by Campus France (to EG), and from the French Minister for Higher Education, Research and Innovation (to MB). The mass spectrometry instrumentation was funded by the University of Strasbourg, IdEx "Equipement mi-lourd" 2015 (to PH). The funders had no role in the study design, data collection and analysis, decision to publish, or preparation of the manuscript.

**Competing interests:** The authors have declared that no competing interests exist.

## Introduction

In mammalian cells, the main antiviral defense system involves the activation of a signaling cascade relying on production of type I interferon (IFN I). This pathway depends on the recognition of extrinsic signals or pathogen associated molecular patterns (PAMPs) by dedicated host receptors. Double-stranded (ds) RNA, which can originate from viral replication or convergent transcription, is a very potent PAMP and can be sensed in the cell by various proteins among which a specific class of DExD/H-box helicases called RIG-I-like receptors (RLRs) [1]. RLRs comprise RIG-I, MDA5 and LGP2 and transduce viral infection signals to induce expression of IFN I cytokines that act in autocrine and paracrine fashions. These cytokines then trigger the expression of hundreds of interferon-stimulated genes (ISGs) to stop the virus in its tracks [2]. Among those ISGs, dsRNA-activated protein kinase R (PKR) plays an important role in antiviral defense by blocking cellular and viral translation upon direct binding to long dsRNA [3]. PKR is a serine-threonine kinase that dimerizes and auto-phosphorylates upon activation. It then phosphorylates numerous cellular targets among which the translation initiation factor eIF2α, which results in the inhibition of cap-dependent translation [4]. Accordingly, translation of many RNA viruses, including alphaviruses, is inhibited by PKR [5–7]. PKR is also involved in other cellular pathways including apoptosis, autophagy and cell cycle [3,8].

RNAi is another evolutionary conserved pathway triggered by long dsRNA sensing [9]. One key component in this pathway is the type III ribonuclease DICER, which is also essential for micro (mi)RNA biogenesis [10,11]. These small regulatory RNAs are sequentially produced by the two ribonucleases DROSHA and DICER, before being loaded into an Argonaute (AGO) effector protein in order to regulate their target mRNAs [12]. Whatever its substrate, be it long dsRNA or miRNA precursor, DICER relies on interacting with co-factors to be fully functional. In mammalian cells, the TAR-RNA binding protein (TRBP), a dsRNA binding protein (dsRBP), was shown to play a role in the selection of DICER substrates, its stabilization, strand selection and incorporation into AGO2 [13]. The interaction with TRBP is well characterized and depends on the helicase domain of DICER and the third dsRNA binding domain (dsRBD) of TRBP [14]. Another dsRBP, the protein activator of interferon-induced protein kinase R (PACT), was also described as an important cofactor of DICER. Although its function is not fully understood, PACT seems to also participate in miRNA loading and strand selection [15,16] *via* protein-protein interaction between the DICER helicase domain and the third dsRBD of PACT [17].

It is now common knowledge that RNAi is the main antiviral defense system in several phyla such as plants, arthropods and nematodes (reviewed in [18]). However, its exact contribution in the mammalian antiviral response remains unclear [19–21]. Recent studies indicate that a functional antiviral RNAi does exist in mammals in specific cases. An antiviral RNAi response was first detected in undifferentiated mouse embryonic stem cells [22] lacking the IFN response, suggesting that these two pathways could be incompatible. Indeed, in mammalian somatic cells deficient for MAVS or IFNAR, two components of the interferon response, an accumulation of DICER-dependent siRNAs derived from exogenous long dsRNA was detected [23]. In addition, the RLR LGP2 was found interacting with both DICER and TRBP, blocking respectively siRNA production and miRNA maturation [24–26]. Moreover, AGO4 was recently shown to be involved in antiviral RNAi against Influenza A virus (IAV), Vesicular stomatitis virus (VSV) and Encephalomyocarditis virus (EMCV) [27]. Finally, viral suppressors of RNAi (VSRs) have been shown to prevent DICER from playing an antiviral role in mammalian cells [28,29]. Nonetheless, several studies reported no detection of viral siRNAs in mammalian somatic cells infected with several viruses [30–32]. In somatic cells, only a

helicase-truncated form of human DICER could produce siRNAs from IAV genome [33], but it also turned out that these siRNAs cannot confer an antiviral state [34].

Based on these conflicting observations, we decided to study the involvement of DICER during infection of human cells with the Sindbis virus (SINV). SINV is a member of the *Toga-viridae* family in the alphavirus genus, which is transmitted by mosquitoes to mammals and can induce arthritogenic as well as encephalitic diseases [35]. It is widely used as a laboratory alphaviruses model as it infects several cell types and replicates to high titers. SINV has a positive stranded RNA genome of about 12 kb, which codes for two polyproteins that give rise to non-structural and structural proteins, including the capsid. Moreover, upon viral replication, a long dsRNA intermediate, which can be sensed by the host antiviral machinery, accumulates. Of note, SINV dsRNA can be cleaved into siRNAs in insects as well as in human cells expressing the Drosophila DICER-2 protein [36]. Nonetheless, although human DICER has the potential to interact with the viral RNA duplex, we did not find evidence that SINV dsRNA could be processed into siRNAs in somatic mammalian cells [30,36]. We thus hypothesized that specific proteins could interfere with DICER during SINV infection by direct interaction and limit its accessibility and/or activity. To address this hypothesis, we generated HEK293T cells expressing a tagged version of human DICER that could be immunoprecipitated in mock or SINV-infected cells in order to perform a proteomic analysis of its interactome. Among the proteins co-immunoprecipitated with DICER and that were specifically enriched upon infection, we identified dsRBPs such as ADAR1, DHX9, PACT and PKR. We further validated the direct interaction between DICER and PKR upon SINV infection. We also demonstrated that the interactions of the endogenous DICER with PKR, PACT and DHX9 could also be detected in SINV-infected, but not mock-infected, HCT116 cells. We dissected the protein domains necessary for this interaction and we found that DICER helicase domain plays a fundamental role as a recruitment platform for PKR but also for other co-factors. Finally, we also show that expression of a helicase-truncated version of DICER has a negative effect on SINV infection. Importantly, this antiviral phenotype is independent of RNAi, but requires the presence of PKR. Our results indicate that DICER interactome is highly dynamic and directly link components of RNAi and IFN pathways in modulating the cellular response to viral infection.

## Results

### Establishment of a HEK293T cell line expressing FLAG-HA tagged DICER

In order to be able to study the interactome of the human DICER protein during viral infection, we transduced *Dicer* knock-out HEK293T cells (NoDice 2.20) [37] with either a lentiviral construct expressing a FLAG-HA-tagged wild type DICER protein (FHA:DICER WT #4) or a construct without insert as a negative control (FHA:ctrl #1). After monoclonal selection of stably transduced cells, we first characterized one clone of both FHA:DICER WT and of the FHA:ctrl cell lines. We first confirmed that the expression of the tagged version of DICER restored the miRNA biogenesis defect observed in the NoDice cells (S1A Fig). We then monitored the phenotype of these cells during SINV infection by using as a readout of viral infection the modified version of SINV able to express GFP from a duplicated sub-genomic promoter (SINV-GFP) [38]. At 24 hours post-infection (hpi) and a multiplicity of infection (MOI) of 0.02, the GFP fluorescence observed in FHA:DICER WT #4 cells and HEK293T cells was similar. However, the NoDice FHA:ctrl #1 cells displayed a decrease in GFP signal (Fig 1A). Western blot analysis of GFP expression confirmed the observations by epifluorescence microscopy, *i.e.* a significantly lower accumulation of GFP in the absence of the DICER protein (Fig 1B). We therefore wished to confirm the effect of DICER loss on SINV-GFP infection in another NoDice cell line, *i.e.* the NoDice clone 4.25 [39], and in another clone of the NoDice

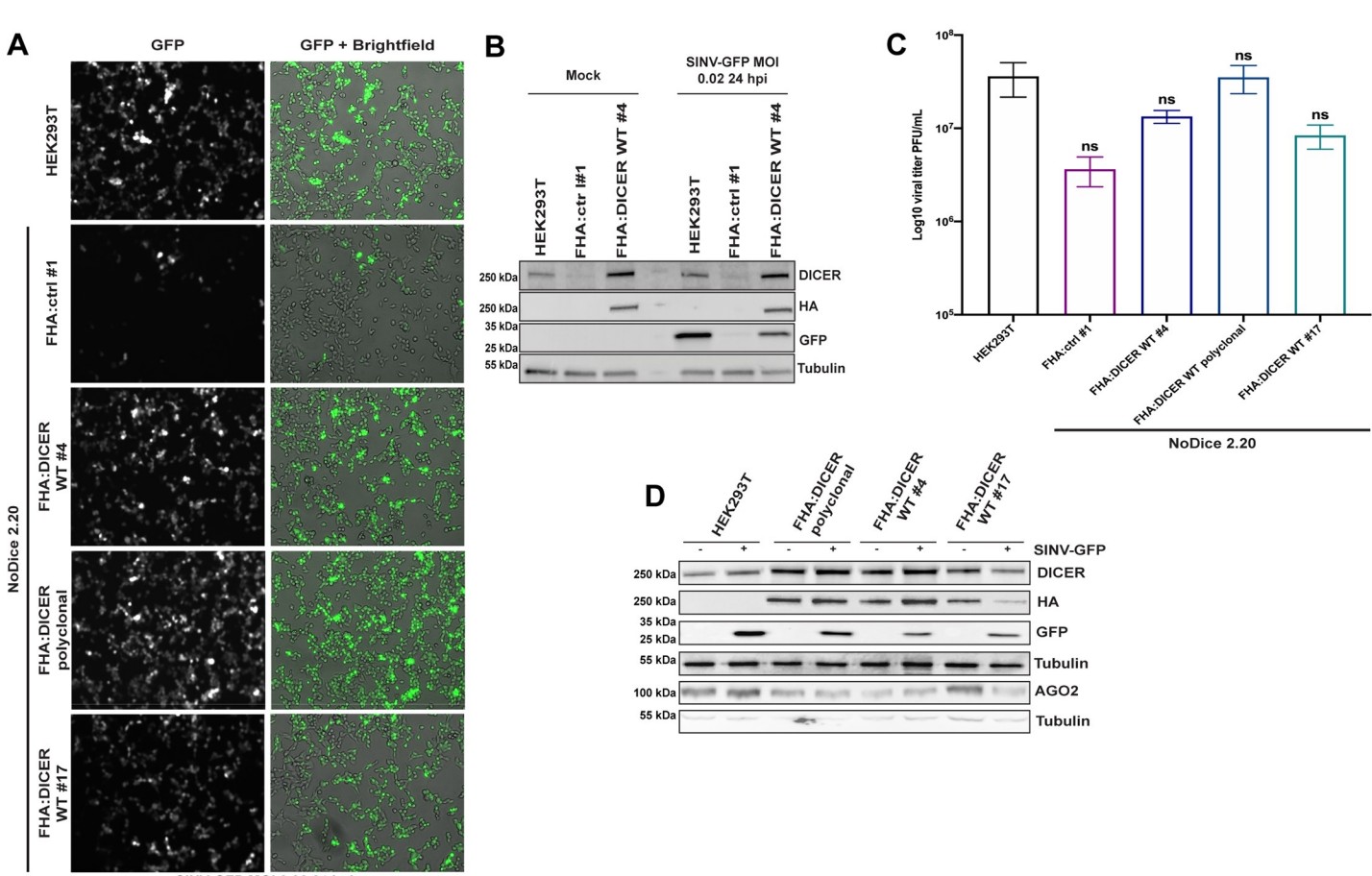

**Fig 1. Analysis of SINV infection in HEK293T cells and characterization of FHA:DICER WT cell lines. A.** GFP fluorescent microscopy pictures of HEK293T, NoDice FHA:ctrl #1 and FHA:DICER cell lines infected (polyclonal and two clones, #4 and #17) with SINV-GFP at an MOI of 0.02 for 24 h. The left panel corresponds to GFP signal from infected cells and the right panel to a merge picture of GFP signal and brightfield. Pictures were taken with a 5x magnification. hpi: hours post-infection. **B.** Western blot analysis of DICER (DICER and HA) and GFP expression in SINV-GFP-infected HEK293T, NoDice FHA:ctrl #1 and FHA:DICER cell lines shown in A. Gamma-Tubulin was used as loading control. **C.** Mean (+/- SEM) of SINV-GFP viral titers in the same cell lines as in A infected at an MOI of 0.02 for 24 h (n = 3) from plaque assay quantification. ns: non-significant, ordinary one-way ANOVA test with Bonferroni correction. **D.** Western blot analysis of DICER (DICER and HA) and AGO2 expression in HEK293T, NoDice FHA:ctrl #1 and FHA:DICER cell lines. Gamma-Tubulin was used as loading control.

2.20 FHA:ctrl cells (NoDice FHA:ctrl #2). We observed a similar decrease of SINV-GFP infection in NoDice 2.20 cells and two independent NoDice FHA:ctrl clones compared to HEK293T cells as shown by GFP microscopy (S1B Fig), by titration of the virus (S1C Fig) and by western blot analysis (S1D Fig). However, the independent NoDice 4.25 *Dicer* knock-out clone appeared mostly unaffected compared to HEK293T cells in term of GFP accumulation and viral titer (S1B, S1C and S1D Fig). This suggests that, despite the observed slight effect on SINV-GFP in NoDice 2.20 cells (Fig 1), DICER proviral effect is not reproducible in an independent clone and therefore could not be generalized.

In order to evaluate whether different expression levels of DICER in a NoDice background could rescue the SINV infection phenotype observed in HEK293T cells, we also infected both the FHA:DICER WT polyclonal and an independent FHA:DICER WT clone (FHA:DICER WT #17) with SINV-GFP (Fig 1A, 1C and 1D). We confirmed that the GFP fluorescence observed by microscopy (Fig 1A), as well as the viral titers and the GFP protein accumulation (Fig 1C and 1D) in all tested FHA:DICER lines were comparable to the ones observed in

HEK293T cells. Moreover, there was no striking difference in AGO2 expression between the FHA:DICER lines (Fig 1D).

Altogether, these results indicate that the FHA-tagged DICER protein can functionally complement the lack of DICER in terms of miRNA biogenesis (S1A Fig) and can therefore be used for proteomics studies. Moreover, because we could not observe significant differences in terms of SINV infection (Fig 1) between the different FHA:DICER clones tested, we decided to select one line, namely FHA:DICER WT #4, for further analysis.

## Analysis of DICER interactome during SINV infection by mass spectrometry

Our molecular tool being validated, we then focused on determining the interactome of FHA: DICER during SINV infection. We wanted to look at DICER interactome at an early infection time point to isolate cellular factors that could potentially modulate either DICER accessibility or its effect on viral dsRNA. As SINV replicates quickly upon cellular entry, we chose to set up the infection conditions to a duration of 6 hours at an MOI of 2.

We performed an anti-HA immunoprecipitation experiment (HA IP) coupled to label-free LC-MS/MS analysis in FHA:DICER WT #4 cells either mock-infected or infected for 6 h at an MOI of 2 with SINV-GFP. In parallel, we performed an anti-MYC immunoprecipitation as a negative control (CTL IP). The experiments were performed in technical triplicate in order to have statistically reproducible data for the differential analysis, which was performed using spectral counts. Prior to the detailed analysis of the results, we verified that there was no confounding factor in the experimentation by performing a Principal Component Analysis (PCA). This allowed us to see that the replicates were very homogenous and that the different samples were well separated based on the conditions.

To check the specificity of the HA immunoprecipitation, we first compared the proteins identified in the HA IP with the ones identified in the CTL IP in mock-infected cells. Differential expression analysis allowed us to calculate a fold change and an adjusted p-value for each protein identified and to generate a volcano plot representing the differences between HA and CTL IP samples. Applying a fold change threshold of 2 (abs(LogFC)>1)), an adjusted p-value threshold of 0.05 and a cutoff of at least 5 spectral counts in the most abundant condition, we identified 258 proteins differentially immunoprecipitated between the two conditions out of 1318 proteins (Fig 2A and S1 Table). Among these, 123 proteins were specifically enriched in the HA IP. The most enriched protein was DICER, followed by its known co-factors TRBP and PACT (also known as PRKRA) [13,17]. We were also able to retrieve AGO2, indicating that the RISC loading complex was immunoprecipitated and that proteins retrieved in our HA IP are specific to DICER immunoprecipitation.

We next performed the differential expression analysis of proteins retrieved in the HA IP in SINV-GFP compared to mock-infected cells. Among 1342 proteins, 296 were differentially retrieved between conditions (Fig 2B and S2 Table). Of these, 184 proteins, including viral ones, were at least 2-fold enriched in SINV-GFP-infected cells. GO-term analysis showed a significant enrichment in RNA binding proteins including double-stranded RNA binding proteins and RNA helicases (Fig 2C). We then generated a functional protein association network using STRING on the top 100 proteins enriched in SINV-infected compared to mock-infected cells (Fig 2D). The resulting STRING network confirmed that a limited number of these proteins are known to be interacting with DICER, but that they are all engaged in other complexes (*e.g.* DHX9, DDX18) that could partly explain the presence of some candidates in the mass spectrometry data. In addition, a large number of these proteins are involved in RNA metabolic processes (Fig 2D, in red), or in their regulation (Fig 2D, in blue), while a whole cluster is

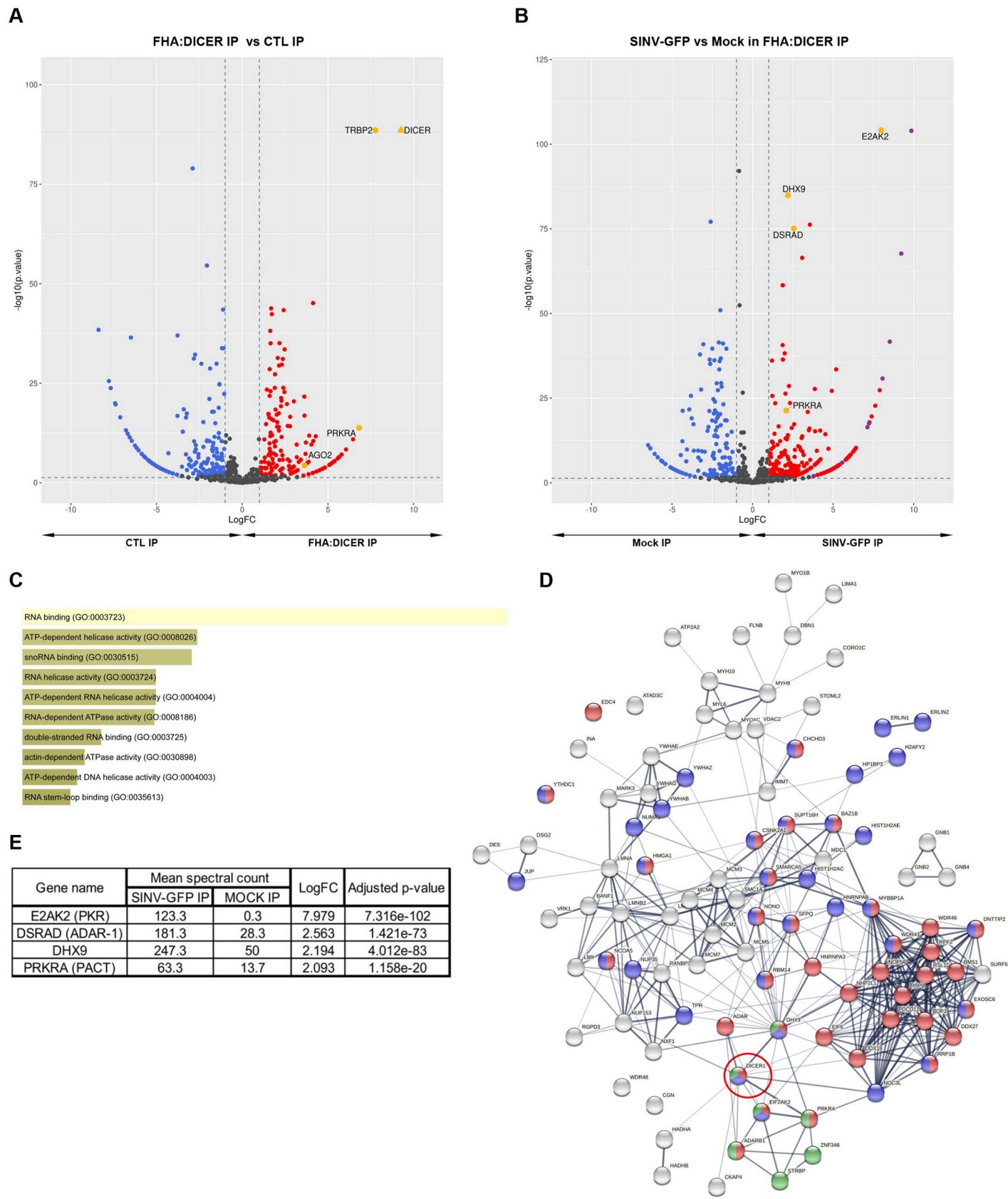

**Fig 2. LC-MS/MS analysis of DICER interactome during SINV infection. A.** Volcano plot for differentially expressed proteins (DEPs) between HA IP and CTL IP in FHA:DICER mock-infected cells. Each protein is marked as a dot; proteins that are significantly up-regulated in HA IP are shown in red, up-regulated proteins in CTL IP are shown in blue, and non-significant proteins are in black. The horizontal line denotes a p-value of 0.05 and the vertical lines the Log2 fold change cutoff (-1 and 1). DICER and its cofactors (TRBP, PACT, AGO2) are highlighted in yellow. **B.** Volcano plot for DEPs between SINV-GFP (MOI of 2, 6 hpi) and mock fractions of HA IP in FHA:DICER cells. Same colour code and thresholds as in A have been applied. Proteins that are discussed in the text are highlighted in yellow and SINV proteins in purple. **C.** Gene Ontology (GO) term enrichment of proteins up-regulated in SINV-GFP fraction of HA IP using Enrichr software [89,90]. The graph displays the GO term hierarchy within the "biological process" branch sorted by p-value ranking computed from the Fisher exact test. The length of each bar represents the significance of that specific term. In addition, the brighter the colour is, the more significant that term is. Viral proteins have been excluded for this analysis. **D.** STRING interaction network of the top 100 proteins enriched in SINV-infected vs. mock-infected cells. Proteins involved in RNA metabolic processes or the regulation thereof are indicated in red and blue respectively, proteins with a known dsRNA binding function are indicated in green. DICER is indicated by a red circle. **E.** Summary of the differential expression analysis of SINV-GFP vs mock fractions from HA IP in FHA:DICER cells. The analysis has been performed using a generalized linear model of a negative-binomial distribution and p-values were corrected for multiple testing using the Benjamini-Hochberg method.

composed of dsRNA binding proteins (Fig 2D, in green). Among the RNA binding proteins retrieved, the top and most specific DICER interactor is the interferon-induced, double-stranded (ds) RNA-activated protein kinase PKR (also known as E2AK2), which is enriched more than 250 times in virus-infected cells (Fig 2B and 2E). We were also able to identify the dsRNA-specific adenosine deaminase protein ADAR-1 (also known as DSRAD), as well as PACT, which were enriched 5.9 and 4.2 times respectively in SINV-GFP-infected cells compared to mock-infected cells (Fig 2B and 2E). Among the isolated RNA helicases, we identified the ATP-dependent RNA helicase A protein DHX9, which is implicated in *Alu* element-derived dsRNA regulation and in RISC loading [40,41]. In order to verify if the observed interactions were specific to SINV we performed the same experiments with another virus of the *Togaviridae* family, the Semliki forest virus (SFV). In this analysis, we were able to retrieve ADAR-1, DHX9, PACT and PKR, specifically enriched in SFV-infected samples (S2 Fig and S3 and S4 Tables). These results show that these interactions can be retrieved in *Togaviridae*-infected cells.

Taken together, our data indicate that several proteins interacting with DICER in virus-infected cells are involved in dsRNA sensing and/or interferon-induced antiviral response.

## DICER and PKR interact *in vivo* in the cytoplasm during SINV infection

To validate the LC-MS/MS analysis, we performed a co-immunoprecipitation (co-IP) followed by western blot analysis in FHA:DICER WT #4 cells infected with SINV-GFP at an MOI of 2 for 6 h. Whereas TRBP interacted equally well with FHA:DICER in mock and SINV-GFP-infected cells, ADAR-1, PKR, DHX9 and PACT were only retrieved in the HA IP in SINV-GFP-infected cells (Fig 3A). We verified that these interactions could also be observed at a later time post-infection by performing the HA IP in FHA:DICER WT #4 cells infected with SINV-GFP for 24 h at an MOI of 0.02. This indicates that the specific interactions between DICER and ADAR-1, DHX9, PACT or PKR occur at an early stage of the SINV infection and remain stable in time in virus-infected cells (S3A Fig).

In order to verify whether these interactions were mediated by RNA, we performed an anti-HA co-IP experiment on an RNase A/T1 treated total extract from FHA:DICER WT #4 cells infected with SINV-GFP at an MOI of 2 for 6 h. Since the RNase treatment was performed at relatively low salt concentration (140 mM NaCl), RNase A should cleave dsRNA [42,43] and we should therefore assess both ss and dsRNA-dependency in these conditions. We confirmed the efficiency of the RNase treatment by ethidium bromide staining visualisation of total RNA on an agarose gel (S3B Fig). TRBP equally interacted with FHA:DICER, with or without RNase treatment, in mock and SINV-GFP-infected cells (Fig 3B). Instead, the virus-induced interactions between DICER and PKR or PACT upon SINV-GFP infection were almost totally lost in the RNase-treated samples. Upon virus infection, PKR is phosphorylated to be activated and exert its antiviral function [4]. Using an antibody targeting the phosphorylated form of

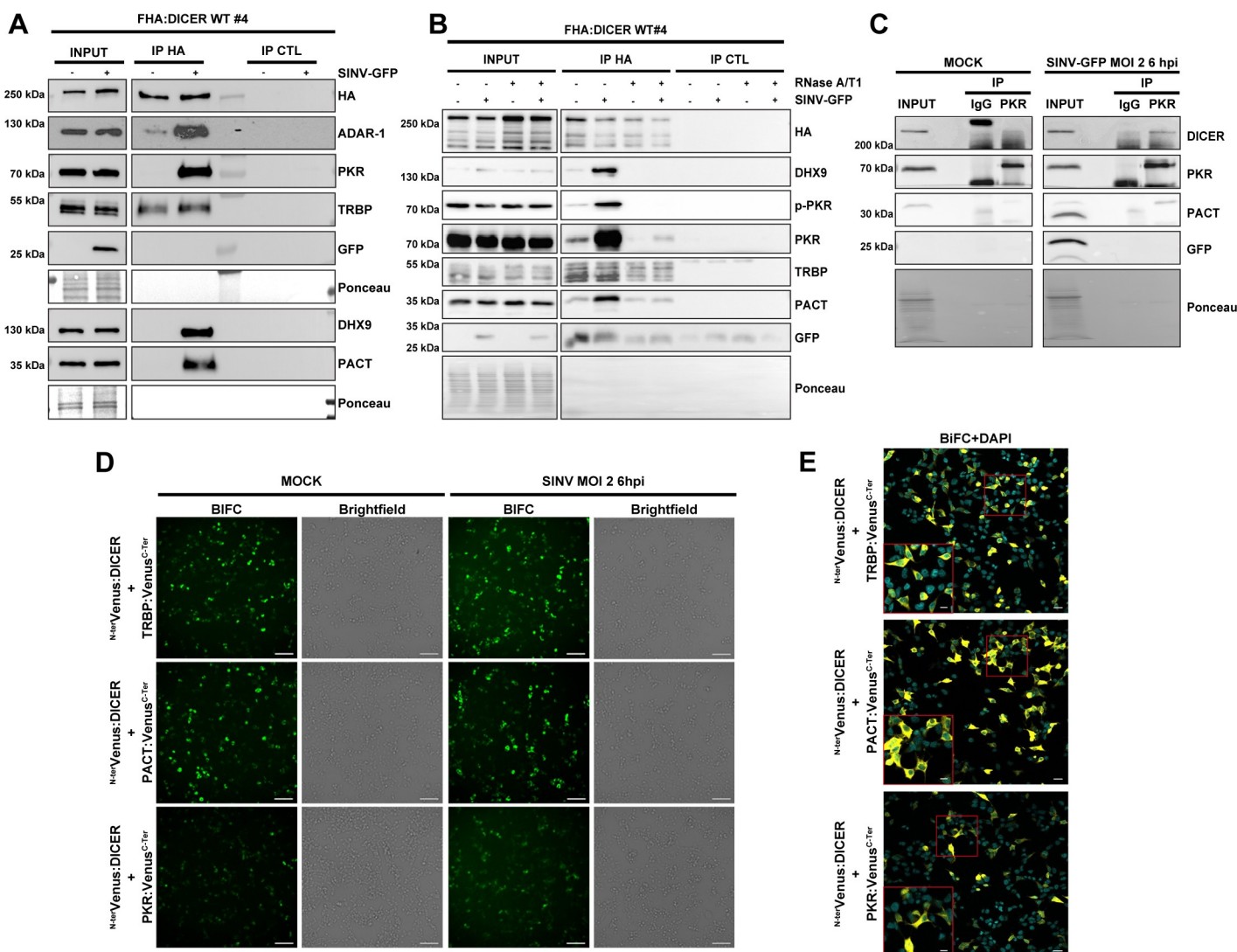

**Fig 3. Confirmation of LC-MS/MS analysis by co-IP and BiFC. A.** Western blot analysis of HA co-IP in mock or SINV-GFP-infected (MOI of 2, 6 hpi) FHA:DICER WT #4 cells. Proteins associated to FHA:DICER were revealed by using antibodies targeting endogenous ADAR-1, PKR, TRBP, DHX9 or PACT proteins. In parallel, an HA antibody was used to verify the IP efficiency and GFP antibody was used to verify the infection. Ponceau was used as loading control. **B.** Western blot analysis of HA co-IP in mock or SINV-GFP-infected (MOI of 2, 6 hpi) FHA:DICER WT #4 cells. The lysate was treated or not with RNase A/T1. Proteins associated to FHA: DICER were revealed by using antibodies targeting endogenous DHX9, p-PKR, PKR, TRBP, or PACT proteins. In parallel, an HA antibody was used to verify the IP efficiency and GFP antibody was used to verify the infection. Ponceau was used as loading control. **C.** Western blot analysis to validate the interaction of PKR with DICER (upper panel) and PACT (lower panel) in mock or SINV-GFP-infected HEK293T cells (MOI of 2, 6 hpi). Immunoprecipitated proteins obtained from PKR pulldowns were compared to rabbit IgG pulldowns to verify the specificity of the assay. **D.** Interactions between DICER and TRBP, PACT or PKR were visualized by BiFC. Plasmids expressing $^{N-ter}$Venus:DICER and TRBP:, PACT: or PKR:Venus$^{C-ter}$ were co-transfected in NoDiceΔPKR cells for 24 h and cells were either infected with SINV at an MOI of 2 for 6 h or not. The different combinations are indicated on the left side. Reconstitution of Venus (BiFC) signal was observed under epifluorescence microscope. For each condition, the left panel corresponds to Venus signal and the right panel to the corresponding brightfield pictures. Scale bar: 100 μm. hpi: hours post-infection. **E.** BiFC experiment on fixed NoDiceΔPKR cells treated as in D. After fixation, cells were stained with DAPI and observed under confocal microscope. Only a merge picture of BiFC and DAPI signals of SINV-infected cells is shown here. A higher magnification of picture showing cytoplasmic localization of the interaction represented by a red square is shown in the bottom left corner. Scale bars: 20 μm and 10 μm.

PKR (p-PKR), we looked for p-PKR before and after RNase treatment. The virus-enriched interactions between DICER and p-PKR or DHX9 were completely lost upon RNase treatment. These results therefore indicate that RNA molecules (either single- or double-stranded)

facilitate DICER interaction with DHX9, PACT and PKR and its active form, although the complex may also partially interact in an RNA-independent manner.

Because of the involvement of PKR in antiviral response [44] and the fact that it shares common co-factors with DICER, namely TRBP and PACT [45,46], we decided to focus our analysis on the DICER-PKR interaction. To confirm the biological relevance of this interaction, we first performed a reverse co-IP to immunoprecipitate the endogenous PKR protein in HEK293T cells infected or not with SINV-GFP. While PACT interacted with PKR both in mock and in SINV-GFP-infected cells as expected (Fig 3C), DICER co-immunoprecipitated with the endogenous PKR only in virus-infected cells thereby confirming the specificity of the interaction between the two proteins (Fig 3C).

To further determine whether DICER and PKR could directly interact *in vivo*, we set up a bi-molecular fluorescent complementation assay (BiFC) experiment [47]. To this end, we fused the N- or C-terminal half of the Venus protein (N-terVenus or C-terVenus) to DICER and to PKR but also to TRBP and PACT. Since we showed above that an N-terminally tagged DICER was functional, we fused the Venus fragments at the N-terminal end of DICER. For the other three proteins, we fused the Venus fragments at the N- or C-terminus and selected the best combination. To avoid interaction with the endogenous DICER and PKR proteins, we conducted all BiFC experiments in NoDiceΔPKR HEK293T cells [33]. In order to control the BiFC experiments, we chose to exploit the well characterized DICER-TRBP interaction, which is known to occur via the DICER DEAD-box helicase domain [14]. We therefore used the wild-type DICER protein as a positive control and a truncated version of DICER protein lacking part of this helicase domain and called DICER N1 [33] as a negative control (S3C Fig). We first confirmed the expression of the tagged proteins by western blot analysis (S3D Fig) and then, we tested the interactions between DICER and TRBP or PACT or PKR. We co-transfected the Venus constructs for 24 h and then infected cells with SINV or not for 6 h at a MOI of 2. A comparable fluorescent signal was observed both in mock- and SINV-infected cells when N-terVenus:DICER was co-transfected with either PACT or TRBP fusion construct (Fig 3D). Although we initially expected an increase of the Venus fluorescence in SINV-infected cells, overall we observed a similar signal for the DICER-PKR interaction both in mock- and SINV-infected cells, probably due to the fact that both proteins are transiently overexpressed in this experiment. The same holds true for the DICER-PACT interaction that can also be seen both in mock- and SINV-infected cells.

As a control and to rule out any aspecific interactions between the different proteins tested, we also monitored the DICER-N1-TRBP interaction by BiFC. As expected, no fluorescent signal was observed in cells co-transfected with N-terVenus:DICER N1 and TRBP:VenusC-ter (S3E Fig), confirming that DICER helicase domain is required for its interaction with TRBP [14] and validating the specificity of the BiFC approach.

To further confirm that the absence of PKR did not influence the interactions of TRBP or PACT with DICER, we also performed a BiFC analysis in HEK293T cells. After verifying that in this context as well, fusion proteins were expressed as expected (S3F Fig), we observed that the results were similar as in NoDiceΔPKR cells (S3G Fig).

To gain more insight into the subcellular localization of these interactions during SINV infection, we performed the BiFC experiments, fixed the cells and observed them under a confocal microscope. We observed a cytoplasmic fluorescent signal for DICER-TRBP and DICER-PACT interactions (Fig 3E upper and middle panels), which is in agreement with their canonical localization for the maturation of miRNAs [10,48]. Similarly, co-transfection of DICER and PKR led to a strong Venus signal homogeneously distributed in the cytoplasm (Fig 3E lower panel).

Collectively, these results formally confirm that DICER interacts with several RNA helicases and dsRNA-binding proteins in virus-infected cells, among which PKR, and that for the latter this interaction occurs in the cytoplasm.

## DICER interactome changes upon SINV infection are not cell-type specific

To further validate our DICER interactome results and generalize them to another biological system, we performed co-IP experiments on the endogenous DICER in a different cell type. To this end, the FLAG-HA-GFP tag was knocked into (KI) the *Dicer* locus in human colon carcinoma cells (HCT116) by CRISPR-Cas9-mediated homologous recombination (S4A, S4B and S4C Fig). A guide RNA (gRNA) targeting the region corresponding to Dicer ATG and a DNA template for homologous recombination bearing the FLAG-HA-GFP sequence surrounded by the upstream and downstream arms of Dicer were used to generate the resulting cell line referred to as HCT116 KI-DICER cells. The expected insertion of the tag in one of the two Dicer alleles was assessed by PCR amplification and Sanger sequencing (S4A, S4B and S4C Fig). In agreement, we could detect two bands for DICER protein by western blot in the HCT116 KI-DICER cells, which confirmed that this cell line is heterozygous (Fig 4A).

We additionally verified the expression of specific DICER-interacting proteins, such as AGO2, PKR or TRBP, in HCT116 KI-DICER cells compared to the parental HCT116 cells and to HEK293T cells (Fig 4A). We also measured the production of mature miRNAs, such as miR-16, by northern blot analysis and confirmed that miRNA expression is maintained in HCT116 KI-DICER cells (Fig 4B). Of note, the GFP inserted at the *Dicer* locus could not be detected by epifluorescence microscopy in the HCT116 KI-DICER cells, which probably reflects the low abundance of the DICER protein.

We then determined whether SINV-GFP infection was comparable in HCT116 cells and HEK293T cells. We infected HCT116, HCT116 KI-DICER and HEK293T cells with SINV-GFP at three different MOI (0.02, 0.1 and 1) and measured GFP fluorescence by microscopy at 24 hpi (Fig 4C). Both HCT116 and HCT116 KI-DICER cells expressed GFP upon infection with SINV-GFP, although with a lower intensity than HEK293T cells. We also verified by western blot analysis the accumulation of GFP and the phosphorylation of both PKR and eIF2α upon SINV-GFP infection of HCT116 KI-DICER and HEK293T cells (S4D Fig) and chose as optimal SINV-GFP condition of infection in HCT116 KI-DICER cells the MOI of 0.1 for 24 h.

To validate the DICER interactions observed in HEK293T cells. We then performed anti-HA co-IP experiments followed by western blot analysis in HCT116 KI-DICER cells infected or not with SINV-GFP. We successfully retrieved TRBP interacting with DICER in both mock and infected cells, whereas DHX9, PKR (phosphorylated or not) and PACT were only retrieved in the HA IP in infected cells (Fig 4D). These results not only confirm that the endogenous DICER specifically interacts with DHX9, PACT and PKR upon SINV infection, but also that these interactions are not restricted to one specific cell type.

## The helicase domain of DICER is required for its interaction with PKR

Even though DICER and PKR are likely brought together by RNA, specific protein domains might be involved in stabilizing the complex. Therefore, we next determined the domain of DICER required for its interaction with PKR. Since its helicase domain was previously shown to be involved in the interaction with TRBP and PACT [14,17], we speculated that it could also be implicated in binding PKR. To test this hypothesis, we cloned several versions of DICER proteins wholly or partly deleted of the helicase domain (Fig 5A DICER N1 and N3). In addition, we also cloned the helicase domain alone (Fig 5A DICER Hel.) and a DICER variant

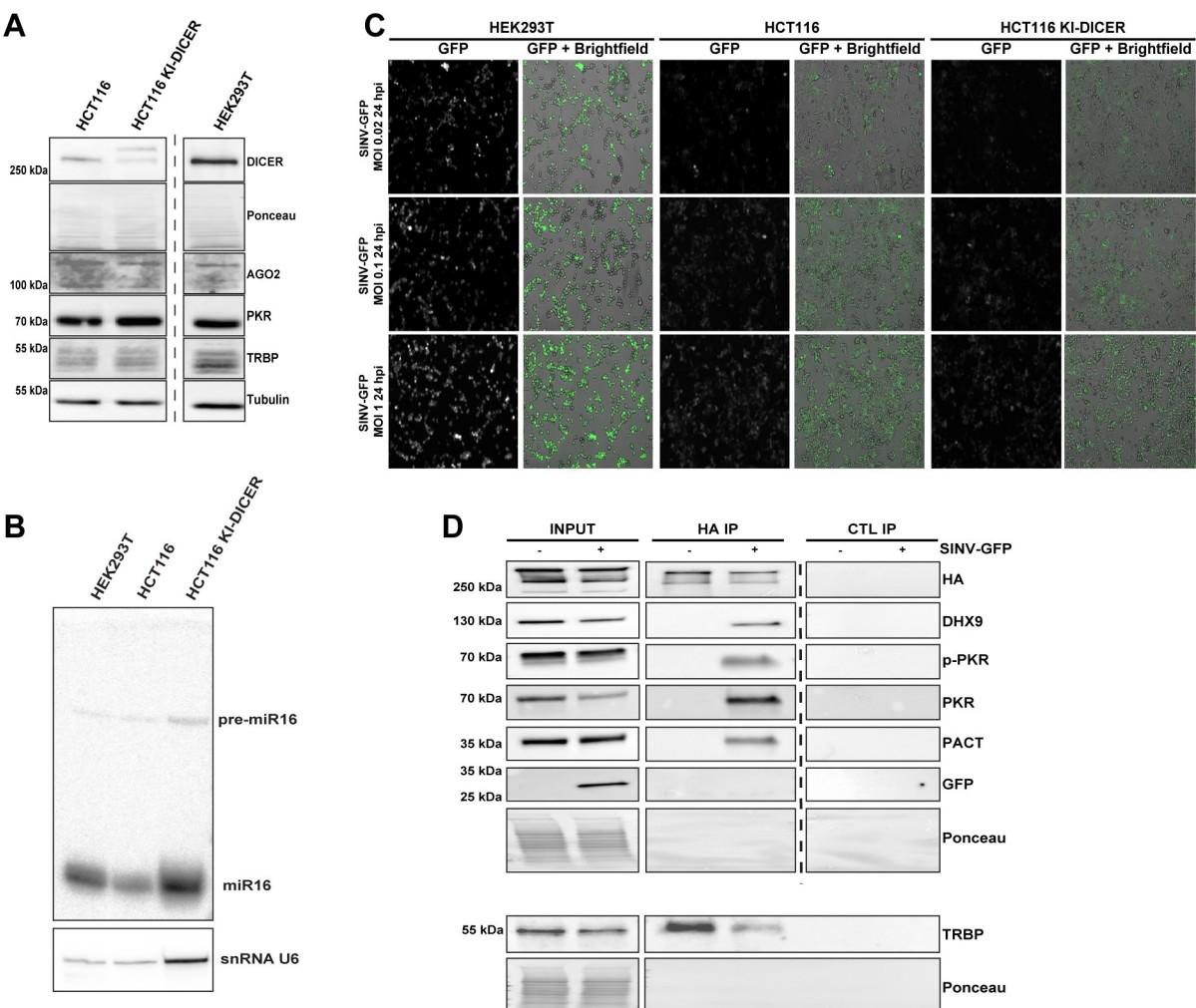

**Fig 4. Confirmation of DICER interactome upon SINV infection in HCT116 KI-DICER cells. A.** Western blot analysis of DICER, AGO2, PKR and TRBP expression in HEK293T, HCT116 and HCT116 KI-DICER cell lines. Gamma-Tubulin and ponceau were used as loading controls. **B**. GFP fluorescent microscopy pictures of HEK293T, HCT116 and HCT116 KI-DICER cell lines infected with SINV-GFP at an MOI of 0.02, 0.1 and 1 for 24 h. The left panel corresponds to GFP signal from infected cells and the right panel to a merge picture of GFP signal and brightfield. Pictures were taken with a 5x magnification. **C**. miR-16 expression analyzed by northern blot in the same cell lines as in B. Expression of snRNA U6 was used as loading control. **D.** Western blot analysis of HA co-IP in mock or SINV-GFP-infected (MOI of 0.1, 24 hpi) HCT116 KI-DICER cells. Proteins associated to FHA-GFP:DICER were revealed by using antibodies targeting endogenous DHX9, p-PKR, PKR, PACT or TRBP proteins. The TRBP immunoblot was performed by loading the same samples on a separate membrane. In parallel, an HA antibody was used to verify the IP efficiency and GFP antibody was used to verify the infection. Ponceau was used as a loading control.

deleted of its C-terminal dsRNA binding domain (Fig 5A DICER ΔdsRBD) since this domain could also be involved in protein-protein interaction [49,50]. We then transfected the different versions of DICER WT and the deletion mutant constructs in NoDice cells. In mock and SINV-GFP infected cells, whole cell extracts were subjected to anti-HA and anti-MYC (CTL) IP. TRBP was retrieved in both conditions with DICER WT, Hel. and ΔdsRBD (Fig 5B and 5C). In mock cells, PACT and PKR were only found weakly interacting with DICER WT (Fig 5B). In SINV-infected cells, we observed that similar to TRBP and to a lesser extent PACT, N1 and N3 mutations strongly reduced the binding of DICER with PKR (Fig 5C lanes 2–3 and 7–8). Importantly, we also noted that the helicase domain alone could bind PKR, TRBP and

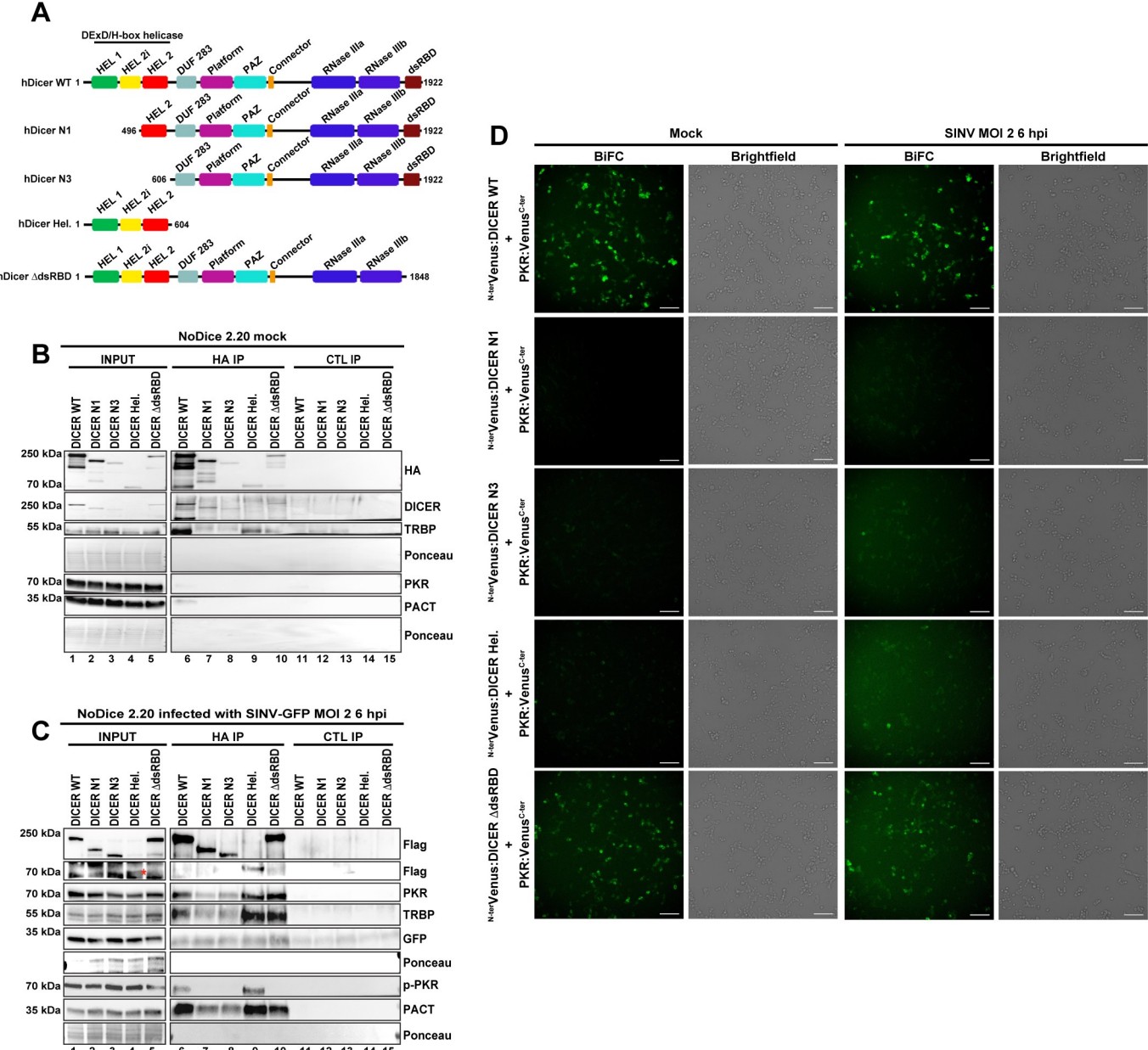

**Fig 5. Identification of DICER domains involved in DICER-PKR interaction. A.** Schematic representation of Human DICER proteins used in this study. The different conserved domains are shown in colored boxes. DUF283: Domain of Unknown Function; PAZ: PIWI ARGONAUTE ZWILLE domain; dsRBD: dsRNA-binding domain. hDICER WT is the full-length protein. hDICER N1 is deleted of the first N-terminal 495 amino acids. hDICER N3 is wholly deleted of the helicase domain. hDICER Hel. is the whole DICER's helicase domain. hDICER ΔdsRBD is deleted of the C-terminal dsRBD. **B.** Western blot analysis of HA co-IP in mock NoDice 2.20 cells transfected with different versions of FHA:DICER proteins. Efficiency of immunoprecipitation was assessed using anti-HA and anti-DICER antibodies and co-IPs of TRBP, PKR and PACT were examined using appropriate antibodies. Expression of GFP in INPUT fraction was visualized as control of SINV-GFP infection. Ponceau staining of membranes is used as loading control. **C.** Western blot analysis of HA co-IP in NoDice 2.20 cells transfected with different versions of FHA:DICER proteins and infected with SINV-GFP (MOI of 2, 6 hpi). Efficiency of immunoprecipitation was assessed using an anti-Flag antibody and co-IPs of PKR, TRBP, p-PKR and PACT were examined using appropriate antibodies. Expression of GFP in INPUT fraction was visualized as control of SINV-GFP infection. Ponceau staining of membranes is used as loading control. The DICER Hel. band is indicated by a red asterisk. **D.** Plasmids expressing the different versions of DICER proteins fused to the N-terminal part of Venus and PKR:Venus^C-ter plasmid were co-transfected in NoDiceΔPKR cells. Cells were treated as in Fig 3D. The different combinations are noted on the left side. The fluorescent signal was observed using an epifluorescence microscope. For each condition, the left panel corresponds to Venus signal and the right panel to the corresponding brightfield pictures. Scale bar: 100 μm. hpi: hours post-infection.

PACT (Fig 5C lanes 4 and 9). Moreover, the deletion of the dsRNA binding domain of DICER did not affect its interaction with TRBP, PACT and PKR (Fig 5C lanes 5 and 10). We also looked for p-PKR in our co-IP (Fig 5C panel p-PKR). We noticed that only WT DICER and its helicase domain were able to interact with p-PKR (Fig 5C lanes 1&6 and 4&9). The fact that DICER ΔdsRBD did not interact with p-PKR (Fig 5C lanes 5&10) is striking but could indicate that the phosphorylation of PKR may induce conformational changes preventing its interaction with some domains of DICER. These results reveal that, like for TRBP and PACT, the helicase domain of DICER is required for DICER-PKR/p-PKR interaction during SINV infection.

In order to confirm these co-IP experiments, we next decided to perform BiFC experiments using the same conditions as previously. In both mock and SINV-infected cells, only the combinations of DICER WT-PKR and DICER ΔdsRBD-PKR showed a strong Venus signal, while neither DICER N1 nor N3 constructs revealed an interaction with PKR (Fig 5D). In contrast, the DICER Hel. construct did not seem to interact with PKR in mock-infected cells but appeared to do so in SINV-infected cells as a faint Venus signal could be observed. These results therefore confirmed the co-IP observations for the DICER-PKR interaction. In addition, we also performed a BiFC experiment using the different DICER constructs with TRBP or PACT. Altogether, the BiFC results mostly fitted with the co-IP experiments for the DICER-TRBP (S5A Fig) and DICER-PACT (S5B Fig) interactions. TRBP indeed did not seem to interact with the DICER N1 and only slightly with the DICER N3. However, PACT interaction was lost with DICER N1, but not with DICER N3 in mock- and SINV-infected cells (S5B Fig third panel). This result may be explained by the fact that DICER interacts with PACT *via* the helicase and DUF domains, whereas only the DICER helicase domain is required for its interaction with TRBP [14,17]. In agreement, the Venus signal observed between the DICER Hel. and PACT seemed weaker than the one we observed with TRBP (S5A and S5B Fig fourth panels).

Taken together these results indicate that DICER interacts with both PKR and its phosphorylated form during SINV infection, and that this interaction requires the helicase domain of DICER.

## Functional importance of DICER helicase domain during SINV infection

We then sought to study the functional role of DICER-PKR interaction during viral infection. For this purpose, we decided to use DICER helicase deletion mutants to study SINV infection. To do so, we first generated NoDice HEK293T cells stably expressing FHA-tagged DICER N1 (FHA:DICER N1) by lentiviral transduction. As for the FHA:DICER WT cell line, we first selected a clone expressing the tagged DICER N1 at a level similar to the endogenous DICER protein in HEK293T cells (Fig 6A). DICER N1 protein has been shown to still be able to produce miRNAs [33]. We thus verified by northern blot analysis that DICER N1 is indeed able to process miRNAs similarly to WT DICER in HEK293T and FHA:DICER cells, thereby validating the functionality of the tagged protein (Fig 6B). We next infected HEK293T, FHA:DICER WT #4 and FHA:DICER N1 #6 cells with SINV-GFP and measured virus accumulation by assessing GFP expression by microscopy analysis. Interestingly, the GFP protein level was drastically reduced in FHA:DICER N1 #6 cells compared to FHA:DICER WT #4 and HEK293T cells (Fig 6C). Encouraged by this observation, we decided to infect with SINV-GFP additional DICER deletion mutants, namely N3 and Hel. We generated stable cell lines for these various mutants by lentiviral transduction in the NoDice 2.20 background and infected those cells with SINV-GFP at an MOI of 0.02 for 24 h. We verified by western blot analysis that all selected DICER mutant clones, namely N1 #6, N3 #2.13 and Hel. #2.6, expressed the tagged protein at the expected size and at levels mostly similar to the FHA:DICER WT #4 cell line (Fig 6D, first two panels). We also verified the DICER mutants contribution to the

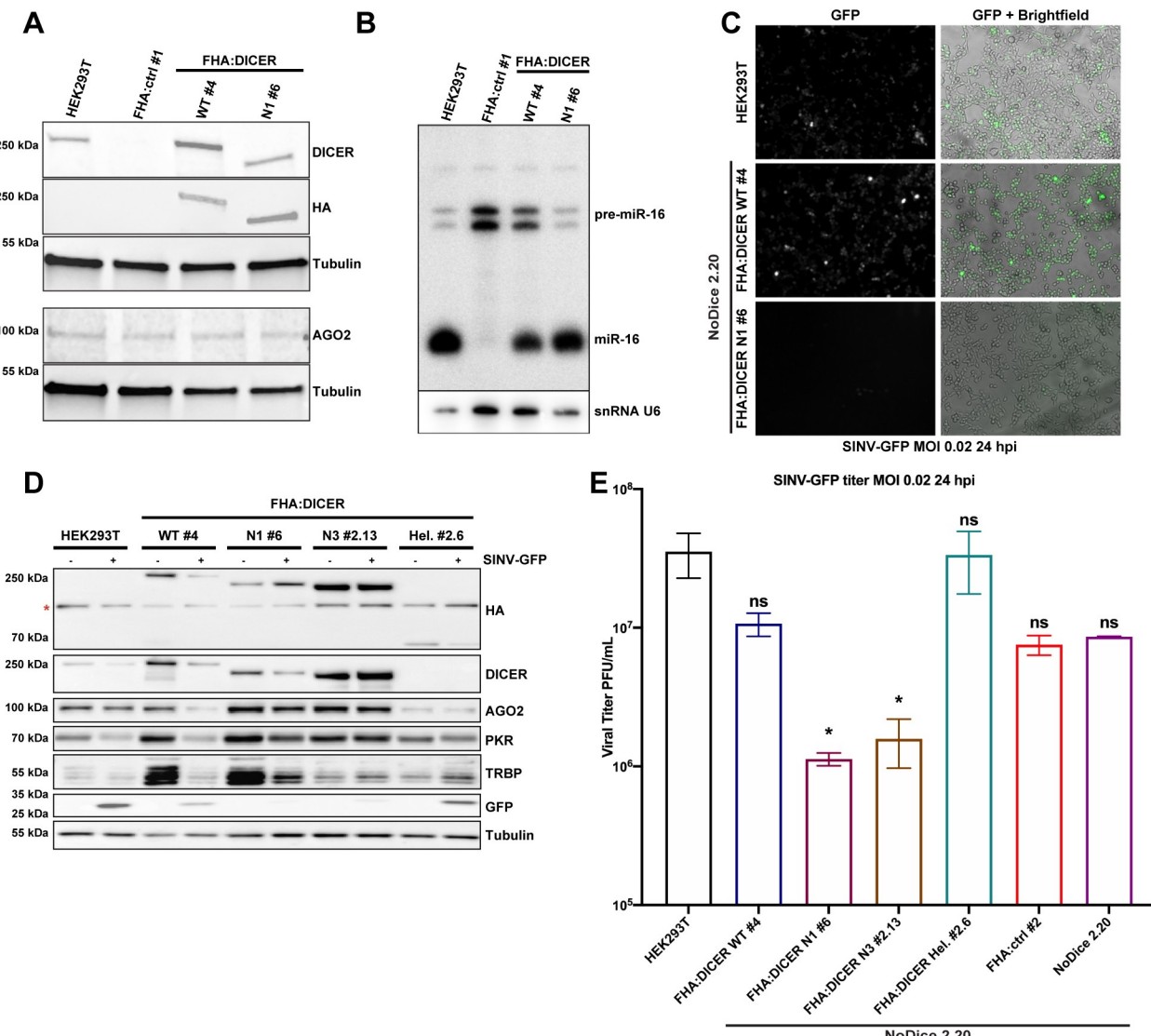

**Fig 6. Analysis of the importance of Dicer helicase domain on SINV-GFP infection in FHA:DICER mutant stable cell lines. A.** Expression of DICER (DICER, HA), TRBP and AGO2 was analysed by western blot in HEK293T, NoDice FHA:ctrl #1, FHA:DICER WT #4 and FHA:DICER N1 #6 cell lines. Gamma-Tubulin was used as loading control. **B.** Northern blot analysis of miR-16 expression in the same samples as in A. Expression of snRNA U6 was used as loading control. **C.** Representative GFP fluorescent microscopy images of HEK293T, FHA:DICER WT #4 and FHA:DICER N1 #6 cell lines infected with SINV-GFP at an MOI of 0.02 for 24 h. The left panel corresponds to GFP signal and the right panel to a merge picture of GFP signal and brightfield. Pictures were taken with a 5x magnification. hpi: hours post-infection. **D.** Western blot analysis of DICER (DICER and HA), AGO2, PKR, and GFP expression in SINV-GFP-infected cells in the same condition as in C. Gamma-Tubulin was used as loading control. The asterisk correspond to aspecific bands **E.** Mean (+/- SEM) of SINV-GFP viral titers fold change over HEK293T cells in HEK293T, NoDice 2.20, FHA:DICER WT #4 and FHA:DICER mutants cell lines infected at an MOI of 0.02 for 24 h (n = 3) from plaque assay quantification. * p < 0.05, ns: non-significant, ordinary one-way ANOVA test with Bonferroni correction.

endogenous miRNA biogenesis by performing a northern blot analysis on miR-16 accumulation (S6A Fig).

We additionally verified the impact of these DICER mutants on SINV-GFP infection by measuring the GFP intensity of fluorescence by microscopy (S6B Fig). Our results indicate that GFP accumulation is similar in HEK293T, NoDice 2.20, FHA:DICER WT, Hel. and ctrl cells. However, almost no fluorescence was detected in FHA:DICER N1 #6 and N3 #2.13 cells compared to HEK293T cells (S6B Fig). The reduction of virus-encoded GFP accumulation

and viral production were confirmed by western blot (Fig 6D) and by plaque assay, respectively (Figs 6E and S6C).

Altogether, these results therefore indicate that expressing a helicase truncated version of DICER, which is unable to interact with PKR, appears to confer an antiviral phenotype against SINV infection.

## The antiviral phenotype of the helicase-truncated DICER mutants is independent of AGO2

We finally carried out a functional analysis of the helicase-domain-truncated DICER N1 and N3 mutants to investigate the mechanism of the antiviral phenotype. First, to investigate a potential implication of the RNAi pathway, we performed a knock-down of the AGO2 protein prior to the infection of NoDice cells expressing either WT, N1 or N3 FHA:DICER. AGO2 is the main effector protein in RNA silencing pathways [51] and has been previously shown to be a crucial antiviral RNAi factor against Influenza A virus in mouse embryonic fibroblasts (MEFs) [52]. We transfected either control siRNAs, or siRNAs targeting AGO2 for 48 h in NoDice cells stably expressing either an empty vector (FHA:ctrl #2) or WT, N1 or N3 FHA:DICER constructs. Cells were then infected with SINV-GFP at an MOI of 0.02 for 24 h, and virus accumulation was first assessed by looking at GFP expression by microscopy analysis (Fig 7A). In all cell lines, no major difference in GFP fluorescence could be observed when comparing cells transfected with the control siRNA or AGO2-specific siRNAs. We verified the knock-down efficiency by western blot analysis and confirmed the microscopy observation by measuring GFP protein accumulation (Fig 7B). Finally, we measured virus accumulation by plaque assay, and we observed that the antiviral phenotype was clearly visible in FHA:DICER N1 #6 and FHA:DICER N3 #2.13 cell lines but was not complemented upon AGO2 knock-down (Fig 7C).

Altogether, these results indicate that the antiviral phenotype against SINV observed in cells expressing helicase-truncated mutant DICER proteins does not depend on the presence of AGO2, thereby ruling out an involvement of RNAi.

## The antiviral phenotype due to the DICER helicase-domain deletion requires PKR

In order to determine the functional role of the PKR-DICER interaction in the antiviral response to SINV, we generated NoDiceΔPKR cells stably expressing either the full length FHA:DICER WT or the helicase deletion mutants FHA:DICER N1 or N3, or the empty vector as a control (FHA:ctrl) by lentiviral transduction. After monoclonal selection of each cell line, we infected them with SINV-GFP at an MOI of 0.02 and assessed virus accumulation by looking at GFP fluorescence by microscopy analysis (Fig 8A). As expected, an increase in GFP fluorescence was observed in NoDiceΔPKR FHA:ctrl cells compared to HEK293T cells at 24 hpi. In contrast we could not observe any difference in GFP fluorescence between NoDiceΔPKR FHA:ctrl cells and those expressing FHA:DICER WT, FHA:DICER N1 or N3 proteins. To verify whether any significant difference in terms of virus accumulation could be observed in NoDiceΔPKR cells expressing WT or helicase truncated DICER proteins, we measured GFP protein levels by western blot analysis (Fig 8B) and virus production by plaque assay (Fig 8C). As opposed to the observations done in NoDice cells expressing PKR (Fig 6), both GFP accumulation and viral titers remained unchanged between NoDiceΔPKR FHA:ctrl cells and those expressing FHA:DICER WT, N1 or N3 constructs. Taken together, these results demonstrate that the antiviral phenotype of helicase-truncated DICER mutants depends on the presence of PKR. Therefore, our data suggest that the helicase domain of DICER sequesters PKR and

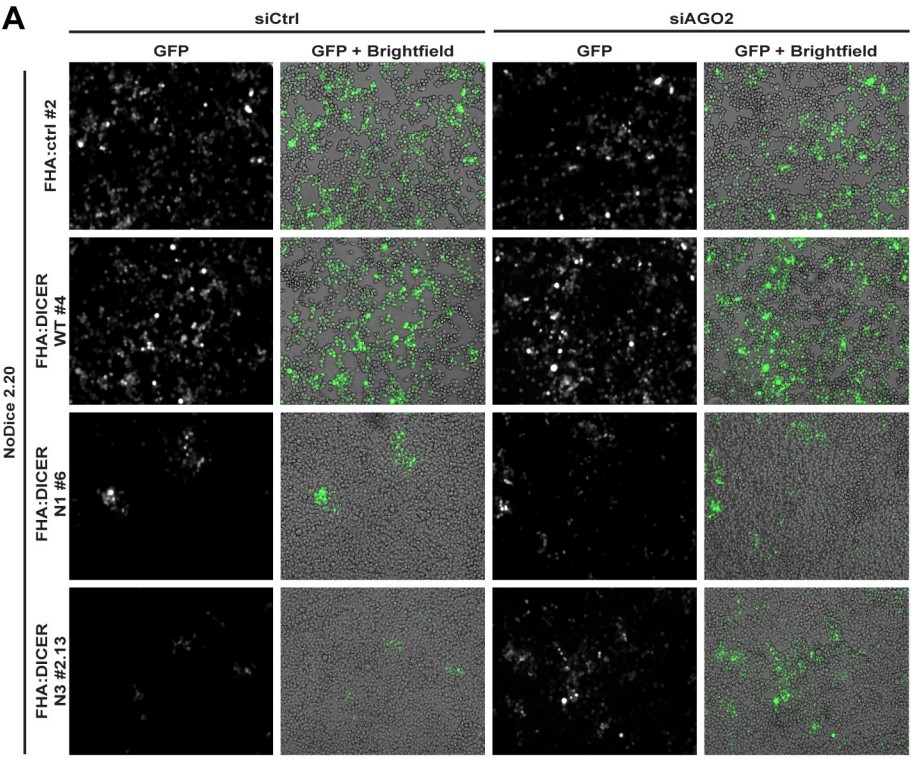

SINV-GFP MOI 0.02 24hpi

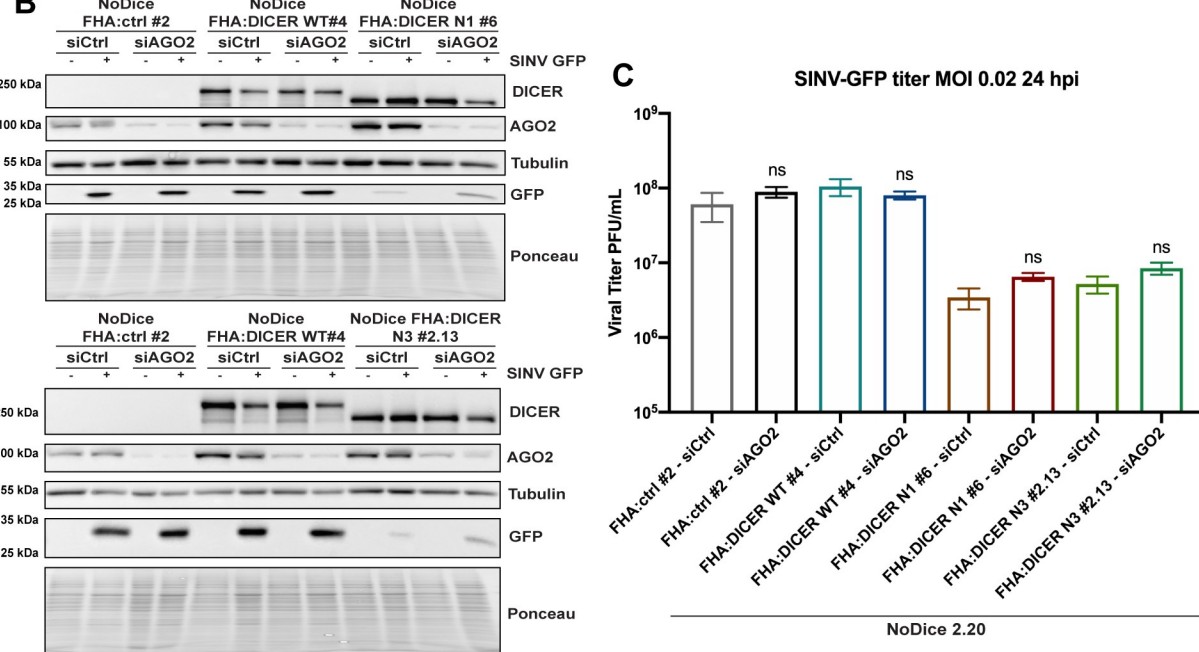

**Fig 7. The antiviral effect of helicase-deleted DICER mutants is independent of AGO2. A.** GFP fluorescent microscopy pictures of NoDice FHA:ctrl #2, NoDice FHA:DICER WT #4 and NoDice FHA:DICER mutant cell lines treated with two doses of siAGO2 at 20 nM for 48 hours before a 24-hour-SINV-GFP infection at an MOI of 0.02. The left panel corresponds to GFP signal from infected cells and the right panel to a merge picture of GFP signal and brightfield. Pictures were taken with 5x magnification. hpi: hours post-infection. **B.** Western blot analysis of DICER, AGO2 and GFP expression in SINV-GFP-infected NoDice FHA:ctrl #2, NoDice FHA:DICER WT #4 and NoDice FHA:DICER mutant cell lines shown in A. Cells were treated with two doses of siAGO2 at 20 nM for 48 hours before a 24-hour-SINV-GFP infection at an MOI of

0.02. Gamma-Tubulin was used as loading control. **C.** Mean (+/-SEM) of SINV-GFP viral titers in the same cell lines as in A. infected at an MOI of 0.02 for 24 h (n = 3) from plaque assay quantification. ns: non-significant, two-tailed unpaired parametric t-test.

when this interaction is lost, the antiviral effect of PKR is exacerbated, thereby explaining the phenotype observed in cells expressing helicase-truncated DICER mutants.

## Discussion

The role of DICER in antiviral defense in human cells remains a topic of intense discussion [21,22,53,54]. In particular there have been contradictory reports regarding its capacity to produce siRNAs from viral RNAs [31,37,55,56]. These observations could be due to the fact that several mammalian viruses potentially encode VSR proteins, thereby masking the effect of RNAi [22,28,29,52,57]. Another putative but non-exclusive explanation could be that there is a

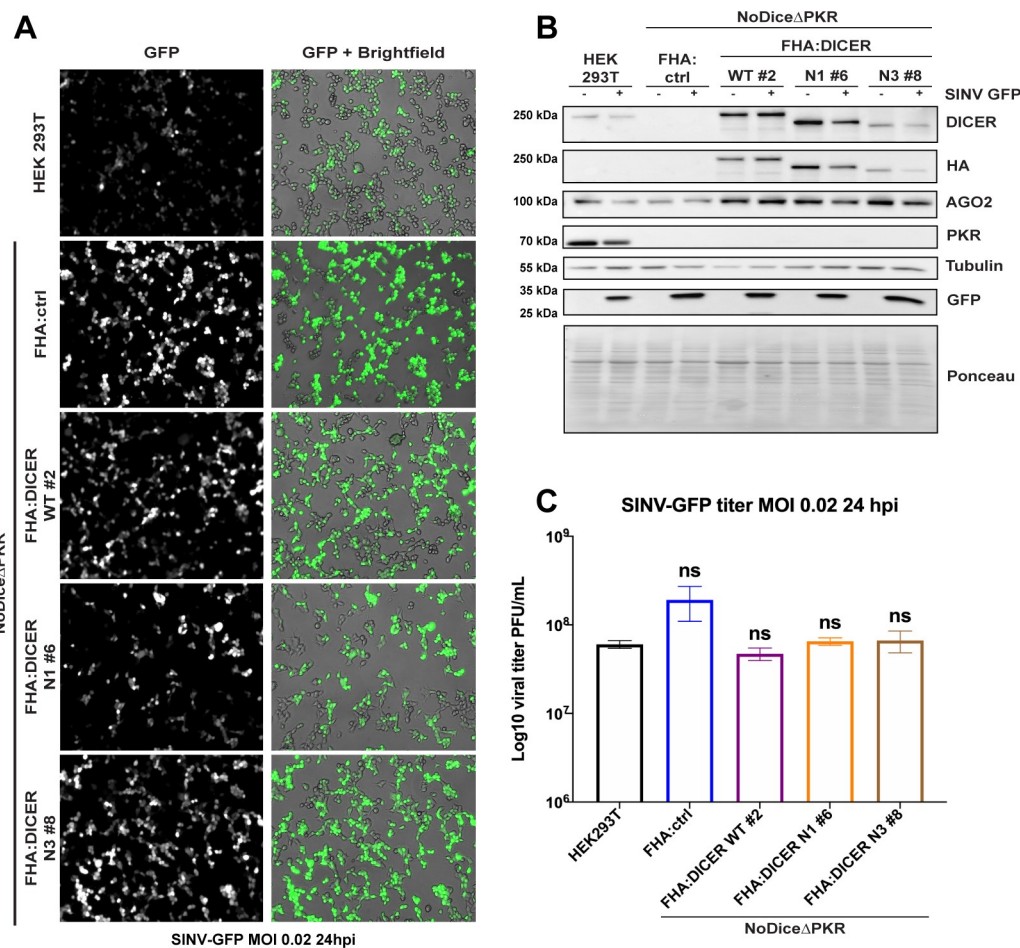

**Fig 8. The antiviral effect of helicase-deleted DICER mutants requires PKR. A.** GFP fluorescent microscopy pictures of HEK293T, NoDiceΔPKR:ctrl and FHA:DICER mutant cell lines infected with SINV-GFP at an MOI of 0.02 for 24 h. The left panel corresponds to GFP signal from infected cells and the right panel to a merge picture of GFP signal and brightfield. Pictures were taken with 5x magnification. hpi: hours post-infection. **B.** Western blot analysis of DICER (DICER and HA), AGO2, PKR and GFP expression in SINV-GFP-infected HEK293T, NoDiceΔPKR FHA:ctrl and FHA: DICER mutant cell lines shown in A. Gamma-Tubulin was used as loading control. **C.** Mean (+/-SEM) of SINV-GFP viral titers in the same cell lines as in A. infected at an MOI of 0.02 for 24 h (n = 3) from plaque assay quantification. ns: non-significant, ordinary one-way ANOVA test with Bonferroni correction.

mutual regulation of type I IFN and RNAi pathways [58,59]. Thus, it has already been shown that PACT can regulate MDA5 and RIG-I during virus infection and therefore the induction of type I IFN response [60,61]. To date, it is not clear whether the activity of the DICER protein as well could be regulated by potential interactors, or inversely whether it could itself modulate the activity of proteins involved in the IFN pathway. To answer this question, we determined the changes in the interactome of human DICER upon SINV and SFV infections. This analysis allowed us to reveal that a lot of proteins associating with DICER during viral infection are dsRNA-binding proteins and RNA helicases. A number of these proteins are known to be involved in antiviral defense pathways, thereby indicating the possible formation of one or several complexes between DICER and these proteins, which are very likely brought together by the accumulation of dsRNA during virus infection.

Among these proteins, we chose to focus on the well-known ISG PKR, which is involved in many cellular pathways such as apoptosis, cellular differentiation, development and antiviral defense [4,8,62,63]. PKR is one of the main actors of the Integrative Stress Response (ISR) in human cells, and its activation or inhibition needs to be tightly regulated in order to have a properly balanced response to stress. Our results indicate that DICER interacts via its helicase domain with PKR in the cytoplasm during SINV infection. The helicase domain of DICER, which is also required for its interaction with TRBP and PACT, belongs to the helicase superfamily 2, which is also found in RLRs such as RIG-I, MDA5 or LGP2 [64,65]. These proteins act as sensors of viral infection and through the activation of proteins such as MAVS, mediate the induction of type I IFN pathway [65]. We hypothesize that even though the human DICER helicase has evolved mainly to act in miRNA/siRNA pathways, it still retained the capacity to act as an RLR. However, as opposed to RIG-I and MDA5, our data suggest that DICER would act more as an inhibitor rather than inducer of the immune response. Therefore, we propose that this domain serves as a platform for the recruitment of different proteins to diversify the functions of DICER.

One such regulatory effect appears to be on the antiviral activity of PKR, as cells expressing a truncated form of DICER unable to interact with PKR become resistant to SINV infection. This is in agreement with previous observations that ectopic expression of the *Drosophila* DICER2 protein in human cells perturbs IFN signaling pathways and antagonizes PKR-mediated antiviral immunity [36]. Although the precise molecular mechanism involved will require further work to be fully deciphered, it seems that the two proteins are likely brought together *via* their interaction with RNA, most probably of viral origin. Indeed, we showed that the co-IP interaction was partially RNase sensitive. However, we confirmed that the interaction is not artificially created during the co-immunoprecipitation procedure, since we could show that DICER and PKR interact in BiFC assay, a technique that favors the detection of direct interactions [47]. Most of the time, the inhibition of PKR activity relies on its inhibition to bind to dsRNA or to auto-phosphorylate. For example, the human tRNA-dihydrouridine synthase 2 (hDus2) binds the first dsRBD of PKR and prevents its activation [66]. TRBP binds dsRNAs but also PKR directly hindering its dimerization. In normal condition, TRBP is also associated with PACT thus preventing PKR activation by PACT [67–70]. Since we showed that DICER can bind the activated phospho-PKR, we hypothesize that this interaction does not result in the inhibition of PKR autophosphorylation. In fact, in condition of infection with a high virus dose, we showed that phospho-PKR levels are similar in cells expressing DICER WT or helicase deletion mutants N1 and N3, but the activated PKR does not associate with these truncated versions of DICER. Therefore, one possibility could be that DICER interaction with PKR prevents the latter from acting upon some of its targets, which remain to be identified, to fine-tune the antiviral response.

As of now, we cannot formally rule out that the effect of DICER on PKR is mediated by other proteins. TRBP and PACT have been shown to regulate PKR activity, the former

normally acting as a repressor and the latter as an activator [46,68,70]. Interestingly, in lymphocytic Jurkat cells infected by HIV-1, PACT can also act as a repressor of PKR [71]. It is thus tempting to speculate that these two proteins participate in the formation of the DICER-PKR complex. However, our results show that this may not necessarily be the case. Indeed, in the BiFC experiment, the DICER N3 mutant still interacted with PACT but not with PKR indicating that PACT binding is not sufficient to confer the association with PKR.

Besides PKR, other proteins were specifically enriched upon viral infection in the DICER IP. These are also interesting candidates to explain the putative regulatory role of DICER. Among these proteins, DHX9 and ADAR-1 are especially intriguing. DHX9, also known as RNA helicase A (RHA), associates with RISC, helping the RISC loading [41]. Moreover, DHX9 is directly involved in removing toxic dsRNAs from the cell to prevent their processing by DICER [40]. It has also been implicated in HIV-1 replication and knockdown of DXH9 leads to the production of less infectious HIV-1 virions [72–74]. Finally, DXH9 interacts with and is phosphorylated by PKR in MEFs. This phosphorylation precludes the association of DHX9 with RNA, thus inhibiting its proviral effect [75]. In light of these observations and ours, we can speculate that the inhibitory effect of DICER on PKR activity could also be linked to DHX9 phosphorylation. ADAR-1 is one of the well-known RNA-editing factors [76]. ADAR-1 is linked to both miRNA biogenesis [77–79] and virus infection. Indeed, ADAR-1 has an antiviral effect against Influenza virus, but most of the time, its depletion leads to a decrease of the viral titer, as was reported for VSV or HIV-1 [80,81]. It has been shown that ADAR-1 and PKR interact directly during HIV-1 infection. This interaction triggers the inhibition of PKR activation, and thus a reduction of eIF2$\alpha$ phosphorylation leading to an increase of virus replication [5,82]. Interestingly, over-expression of ADAR-1 enhances drastically the replication of the alphaviruses Chikungunya virus (CHIKV), and Venezuelan equine encephalitis virus (VEEV) most likely by interfering with the IFN induction [83].

One hypothesis to explain the virus resistance phenotype of the DICER N1 and N3 cell lines could be an increased processivity of these truncated proteins on long dsRNA substrates [33], which would render DICER RNAi proficient. However, our results are not in favor of this hypothesis, since we show that knocking-down AGO2 does not allow to make cells expressing DICER N1 or N3 more sensitive to SINV infection. AGO2 being the only slicer-proficient Argonaute protein expressed at physiological levels in HEK293T cells, we can confidently conclude that the observed phenotype is RNAi-independent.

Finally, we demonstrated that the phenotype of helicase-truncated DICER isoforms depends on PKR expression, because it was completely lost in PKR knockout cell lines. We therefore propose that, at least during infection with SINV, DICER prevents PKR to be fully active by interacting with and potentially sequestrating it. Deciphering the exact molecular mechanism at play will require additional studies in order to get the full picture. Nevertheless, by assessing the interactome of DICER during SINV infection, we have unveiled a new, PKR-dependent, role for the helicase domain of DICER in regulating the cellular response to viral infection.

## Material and methods

### Plasmids, cloning and mutagenesis

Plasmids used for BiFC experiments were a gift from Dr. Oliver Vugrek from the Ruđer Bošković Institute and described in [47]. The cDNAs of TRBP, PACT and PKR were respectively amplified from (pcDNA-TRBP Addgene #15666) [16], (pcDNA-PACT Addgene #15667) [16], (pSB819-PKR-hum Addgene #20030) [84], and cloned into the four pBiFC vectors by Gateway recombination. DICER N1, N3, Hel. and ΔdsRBD were generated by PCR mutagenesis from

pDONR-DICER described in [36] and cloned into the four pBiFC and pDEST-FHA vectors by Gateway recombination. plenti6 FHA-V5 vector was modified from plenti6-V5 gateway vector (Thermo Fisher Scientific V49610) by Gibson cloning. DICER WT, N1, N3 and Hel. from pDONR plasmids were cloned into plenti6 FHA-V5 by Gateway recombination. All primers used are listed in S5 Table.

## Cell lines

HEK293T, HEK293T/NoDice (2.20 and 4.25), and HEK293T/NoDiceΔPKR cell lines were a gift from Pr. Bryan Cullen and described in [33,39]. HCT116 cell line was a gift from Dr. Christian Gaiddon.

## Generation of Flag-HA-GFP-DICER knock-in cell line by CRISPR/Cas9

To generate the knock-in cell line, the sequence of Flag-HA-GFP was amplified by PCR from the Flag-HA-GFP plasmid [85]. DNA sequences corresponding to 1 Kb upstream (left homology arm) and downstream (right homology arm) the starting codon (ATG) of DICER gene were amplified from HCT116 cell genomic DNA using primer pairs listed in S5 Table. The three PCR products were gel-purified and cloned into a linearized pUC19 by In-fusion cloning (Clontech) to obtain the template for homologous recombination (LarmDICER-FlagHAGFP-RarmDICER).

Design of the guide RNA targeting the region between Dicer 5'-UTR and its first coding exon for CRISPR/Cas9 mediated knock-in was carried out using the CRISPOR Design Tool [86]. Annealed oligonucleotides corresponding to the gRNA (S5 Table) were cloned into the vector pX459 (Addgene #48139) which also encodes *S. pyogenes* Cas9 with 2A-Puro.

The sequence of the donor plasmid was additionally mutagenized to disrupt the PAM sequence of the right homology arm to avoid its cleavage by the gRNA.

To obtain the knock-in (KI) cell line, 5 x $10^5$ HCT116 cells were seeded in a 6 well plate with Dulbecco's modified Eagle medium (DMEM, Gibco, Life Technologies) supplemented with 10% fetal bovine serum (FBS, Clontech) in a humidified atmosphere of 5% $CO_2$ at 37˚C and transfected after 24 hours with the pX459-gRNADicerNterm-Cas9-2A-Puro plasmid and the Leftarm-FlagHAGFP-RightarmDICER donor plasmids at the ratio of 1 to 1 (6 micrograms plasmids in total) using Lipofectamine 2000 according to the manufacturer's instructions. 24 hours later, puromycin (1 mg/mL) was added to the cells to increase the KI efficiency and genomic DNA was isolated from individual colonies few days later.

The presence of the Flag-HA-GFP tag in frame with hDICER coding sequence was confirmed by sequencing PCR amplicon from KI cell gDNA. Expression of Flag-HA-GFP N-terminal tagged Dicer protein in the KI cells was confirmed by western blot.

## Cell culture and transfection

Cells were maintained in Dulbecco's modified Eagle medium (DMEM, Gibco, Life Technologies) supplemented with 10% fetal bovine serum (FBS, Clontech) in a humidified atmosphere of 5% $CO_2$ at 37˚C. Transfection was performed using Lipofectamine 2000 (Invitrogen, Thermo Fisher Scientific) according to the manufacturer's instructions.

## Lentivirus production and generation of stable cell lines

The lentiviral supernatant from single transfer vector was produced by transfecting HEK293T cells (ATCC CRL-3216) with 20 μg of the transfer vector, 15 μg of pMDLg/p RRE and 10 μg of pRSV-Rev packaging plasmids (Addgene #12251 and Addgene #12253) and the pVSV

envelope plasmid (Addgene #8454) using Lipofectamine 2000 reagent (Invitrogen, Thermo Fisher Scientific) according to the manufacturer's protocol. Standard DMEM medium (Gibco, Life Technologies) supplemented with 10% Fetal bovine serum (FBS, Gibco, Life Technologies) and 100 U/mL of penicillin-Streptomycin (Gibco, Life Technologies) were used for growing HEK293T cells and for lentivirus production. One 10 cm plate of HEK293T cells at 70–80% confluency was used for the transfection. The medium was replaced 8 hours post-transfection. After 48 hours the medium containing viral particles was collected and filtered through a 0.45 μm PES filter. The supernatant was directly used for transfection or stored at -80˚C. A 6 well plate of HEK293T/NoDice or HEK293T/NoDiceΔPKR cells at 80% confluency was transduced using 600 μL of lentiviral supernatant either expressing FHA:DICER, N1, N3, Hel. or empty vector, supplemented with 4 ug/mL polybrene (Sigma) for 6 hours. The transduction media was then changed with fresh DMEM for 24 hours and the resistant cell clones were selected for about 6 weeks with blasticidin (15 μg/mL for NoDice or 10 μg/mL for NoDiceΔPKR) and subsequently maintained under blasticidin selection.

## Viral stocks, virus infection

Viral stocks of SINV or SINV-GFP were produced as described in [36]. Cells were infected with SINV or SINV-GFP at an MOI of 0.02, 0.1, 1 or 2 and samples were collected at different time points as indicated in the figure legends.

## Analysis of viral titer by plaque assay

Vero R cells were seeded in 96-well plates format and were infected with 10-fold serial dilutions infection supernatants for 1 hour. Afterwards, the inoculum was removed, and cells were cultured in 2.5% carboxymethyl cellulose for 72 hours at 37˚C in a humidified atmosphere of 5% $CO_2$. Plaques were counted manually under the microscope and viral titer was calculated according to the formula: *PFU/mL = #plaques/ (Dilution* Volume of inoculum)*. All data and statistics pertaining to plaque assay analysis can be found in S6 Table.

## Western blot analysis

Proteins were extracted from cells and homogenized in 350 μL of lysis buffer (50 mM Tris-HCl pH 7.5, 150 mM NaCl, 5 mM EDTA, 1% Triton X-100, 0.5% SDS and Protease Inhibitor Cocktail (complete Mini; Sigma Aldrich). Proteins were quantified by the Bradford method and 20 to 30 μg of total protein extract were loaded on 4–20% Mini-PROTEAN TGX Precast Gels (Bio-Rad). After transfer onto nitrocellulose membrane, equal loading was verified by Ponceau staining. For PVDF membrane, equal loading was verified by Coomassie staining after transfer and blotting. Membranes were blocked in 5% milk and probed with the following antibodies: anti-hDicer (1:500, F10 Santa Cruz, sc-136979) and anti-hDicer (1:1000, A301-937A, Bethyl), anti-TRBP (1:500, D-5 Santa Cruz, sc-514124), anti-PKR (1:2500, Abcam ab32506), anti-PACT (1:500, Abcam, ab75749), anti-HA (1:10000, Sigma, H9658), anti-DHX9 (1:500, Abcam, ab26271), anti-p-eIF2 (1:1000, Ser-52 Santa Cruz, sc-601670), anti-hADAR-1 (1:500 Santa Cruz, sc-271854) anti-p-PKR (1:1000 Abcam ab81303) anti-GFP (1:10000, Roche, 11814460001) and anti-Tubulin (1:10000, Sigma, T6557). Detection was performed using Chemiluminescent Substrate (Pierce, Thermo Fisher Scientific) and visualized on a Fusion FX imaging system (Vilber).

## RNA extraction and northern blot analysis

Total RNA was extracted using Tri-Reagent Solution (Fisher Scientific; MRC, Inc) according to the manufacturer's instructions. Northern blotting was performed on 10 μg of total RNA.

RNA was resolved on a 12% urea-acrylamide gel, transferred onto Hybond-NX membrane (GE Healthcare). RNAs were then chemically cross-linked to the membrane during 90 min at 65°C using 1-ethyl-3-[3-dimethylaminopropyl]carbodiimide hydrochloride (EDC) (Sigma Aldrich). Membranes were prehybridized for 30 min in PerfectHyb plus (Sigma Aldrich) at 50°C. Probes consisting of oligodeoxyribonucleotides (see S5 Table) were 5′-end labeled using T4 polynucleotide kinase (Thermo Fisher Scientific) with 25 µCi of [γ-32P]dATP. The labeled probe was hybridized to the blot overnight at 50°C. The blot was then washed twice at 50°C for 20 min (5× SSC/0.1% SDS), followed by an additional wash (1× SSC/0.1% SDS) for 5 min. Northern blots were exposed to phosphorimager plates and scanned using a Bioimager FLA-7000 (Fuji).

## Immunoprecipitation

Immunoprecipitation experiments were carried out either on tagged proteins or on endogenous proteins.

**Tagged proteins.**   Cells were harvested, washed twice with ice-cold 1× PBS (Gibco, Life Technologies), and resuspended in 550 µL of lysis buffer (50 mM Tris-HCl pH 7.5, 140 mM NaCl, 1.5 mM MgCl$_2$, 0.1% NP-40), supplemented with Complete-EDTA-free Protease Inhibitor Cocktail (complete Mini; Sigma Aldrich). Cells were lysed by 30 min incubation on ice and debris were removed by 15 min centrifugation at 2000 g and 4°C. An aliquot of the cleared lysates (50 µL) was kept aside as protein Input. Samples were divided into equal parts (250 µL each) and incubated with 15 µL of magnetic microparticles coated with monoclonal HA or MYC antibodies (MACS purification system, Miltenyi Biotech) at 4°C for 1 hour under rotation (10 rpm). Samples were passed through µ Columns (MACS purification system, Miltenyi Biotech). The µ Columns were then washed 3 times with 200 µL of lysis buffer and 1 time with 100 µL of washing buffer (20 mM Tris-HCl pH 7.5). To elute the immunoprecipitated proteins, 95°C pre-warmed 2x Western blot loading buffer (10% glycerol, 4% SDS, 62.5 mM Tris-HCl pH 6.8, 5% (v/v) 2-β-mercaptoethanol, Bromophenol Blue) was passed through the µ Columns. Proteins were analyzed by western blotting or by mass spectrometry.

**Endogenous proteins.**   mock or SINV-GFP-infected HEK293T cells (MOI of 2) were lysed 6 hours post-infection using immunoprecipitation buffer (50 mM Tris-HCl [pH 7.5], 150 mM NaCl, 5 mM EDTA, 0.05% SDS, 1% triton) supplemented with Complete-EDTA-free Protease Inhibitor Cocktail (complete Mini; Sigma Aldrich). Lysates were treated for 20 min at 37°C with 1 µL of DNase I (Thermo Fisher Scientific) using its buffer (10 mM MgCl$_2$, 5 mM CaCl$_2$ and 1 µL of ribolock). Lysates were cleared by centrifugation at 16000 g, 10 min at 4°C. Supernatants were precleared 1 h at room temperature with magnetic beads blocked with BSA (Thermo Fisher Scientific) to avoid aspecific binding. Lysates were incubated overnight on wheel at 4°C with immunoprecipitation buffer containing magnetic Protein A DynaBeads (Invitrogen, Thermo Fisher Scientific) conjugated with human PKR antibody (Abcam) or negative control rabbit IgG (Cell signaling, Ozyme). Beads were washed 3 times with immunoprecipitation buffer, 3 times with wash buffer (50 mM Tris-HCl [pH 7.5], 200 mM NaCl, 5 mM EDTA, 0.05% SDS, 1% triton, supplemented with Complete-EDTA-free Protease Inhibitor Cocktail (complete Mini; Sigma Aldrich) and twice with cold PBS 1X (Gibco, Life Technologies). Beads were eluted with 2x western blot loading buffer and incubated for 10 min at 95°C under agitation. Proteins were analyzed by western blotting.

## RNase treatment followed by co-IP

*On tagged proteins*: Cells were harvested, washed twice with ice-cold 1× PBS (Gibco, Life Technologies), and resuspended in 550 µL of lysis buffer (50 mM Tris-HCl pH 7.5, 140 mM NaCl,

1.5 mM MgCl$_2$, 0.1% NP-40), supplemented with Complete-EDTA-free Protease Inhibitor Cocktail (complete Mini; Sigma Aldrich). Cells were lysed by 30 min incubation on ice and debris were removed by 15 min centrifugation at 2000 g and 4°C. Lysate was treated or not with RNase A/T1 mix (Thermo Fisher Scientific) and place at 37°C 30 min. An aliquot of the cleared lysates (25 μL) was kept aside as protein Input and another aliquot (25 μL) was kept to assess RNase treatment efficiency. Co-IP was led as previously described.

Total RNA was extracted using Tri-Reagent Solution (Fisher Scientific; MRC, Inc) according to the manufacturer's instructions. RNA integrity upon treatment was verified on an 1% agarose gel containing ethidium bromide 10 mg/mL (Invitrogen, Thermo Fisher Scientific) and revealed under UV on Gel DocEZ system (Bio-Rad).

## siRNA transfection

20 nM of human AGO2 or non-targeting control siRNA (Horizon discovery) were transfected in 130000 NoDice FHA:ctrl #2, NoDice FHA:DICER WT #4, N1 #6 or N3 #2.13 cells using Lipofectamine 2000 transfection reagent (Invitrogen, Thermo Fisher Scientific) according to the manufacturer's instructions. After 24 hours, the cells were again transfected with 20 nM of the same siRNA and incubated overnight. Cells were infected or not with SINV-GFP at an MOI of 0.02 for 24 h. Proteins and supernatants were collected and analyzed by western blotting and plaque assay, respectively.

## BiFC assay

Experiments were carried out in two different ways. For non-fixed cells, NoDiceΔPKR or HEK293T cells were seeded at the density of 1.2 x 10$^5$ cells per well in a 24-well plate. After 16 hours, cells were transfected with equimolar quantities of each plasmid forming BiFC couples. After 24 hours, cells were infected with SINV at an MOI of 2 and pictures were taken 6 hours post-infection using ZOE fluorescent cell imager (Bio-Rad). Proteins were collected with lysis buffer (50 mM Tris-HCl pH 7.5, SDS 0.05%, Triton 1%, 5 mM EDTA, 150 mM NaCl) supplemented with Complete-EDTA-free Protease Inhibitor Cocktail (complete Mini; Sigma Aldrich), and subjected to western blot analysis. For fixed cells, NoDiceΔPKR cells were seeded at the density of 8.10$^4$ cells per well in 8-well Millicell EZ Slides (Merck Millipore), transfected and infected as described previously. At 6 hours post-infection, cells were fixed with 4% formaldehyde and 0.2% glutaraldehyde for 10 min. Cells were then washed with 1× PBS (Gibco, Life Technologies) and stained with 10 μg/μL DAPI (Invitrogen, Thermo Fisher Scientific) in 1× PBS solution (Invitrogen, Thermo Fisher Scientific) for 5 min. Fixed cells were mounted on a glass slide with Fluoromount-G mounting media (Southern Biotech). Images were acquired using confocal LSM780 (Zeiss) inverted microscope with an argon laser (514x nm) and with ×40 immersion oil objective. All pictures obtained from BiFC experiments were treated using FigureJ software (NIH).

## Mass spectrometry analysis

Protein extracts were prepared for mass spectrometry as described in a previous study [87]. Each sample was precipitated with 0.1 M ammonium acetate in 100% methanol, and proteins were resuspended in 50 mM ammonium bicarbonate. After a reduction-alkylation step (dithiothreitol 5 mM–iodoacetamide 10 mM), proteins were digested overnight with sequencing-grade porcine trypsin (1:25, w/w, Promega, Fitchburg, MA, USA). The resulting vacuum-dried peptides were resuspended in water containing 0.1% (v/v) formic acid (solvent A). One sixth of the peptide mixtures were analyzed by nanoLC-MS/MS an Easy-nanoLC-1000 system coupled to a Q-Exactive Plus mass spectrometer (Thermo-Fisher Scientific, USA) operating in

positive mode. Five microliters of each sample were loaded on a C-18 precolumn (75 μm ID × 20 mm nanoViper, 3 μm Acclaim PepMap; Thermo) coupled with the analytical C18 analytical column (75 μm ID × 25 cm nanoViper, 3 μm Acclaim PepMap; Thermo). Peptides were eluted with a 160 min gradient of 0.1% formic acid in acetonitrile at 300 nL/min. The Q-Exactive Plus was operated in data-dependent acquisition mode (DDA) with Xcalibur software (Thermo-Fisher Scientific). Survey MS scans were acquired at a resolution of 70K at 200 m/z (mass range 350–1250), with a maximum injection time of 20 ms and an automatic gain control (AGC) set to 3e6. Up to 10 of the most intense multiply charged ions (≥2) were selected for fragmentation with a maximum injection time of 100 ms, an AGC set at 1e5 and a resolution of 17.5K. A dynamic exclusion time of 20 s was applied during the peak selection process.

### Database search and mass-spectrometry data post-processing

Data were searched against a database containing Human and Viruses UniProtKB sequences with a decoy strategy (GFP, Human and Sindbis Virus SwissProt sequences as well as Semliki Forest Virus SwissProt and TrEMBL sequences (releases from January 2017, 40439 sequences)). Peptides were identified with Mascot algorithm (version 2.3, Matrix Science, London, UK) with the following search parameters: carbamidomethylation of cysteine was set as fixed modification; N-terminal protein acetylation, phosphorylation of serine / threonine / tyrosine and oxidation of methionine were set as variable modifications; tryptic specificity with up to three missed cleavages was used. The mass tolerances in MS and MS/MS were set to 10 ppm and 0.02 Da respectively, and the instrument configuration was specified as "ESI--Trap". The resulting .dat Mascot files were then imported into Proline v1.4 package (http://proline.profiproteomics.fr) for post-processing. Proteins were validated with Mascot pretty rank equal to 1, 1% FDR on both peptide spectrum matches (PSM) and protein sets (based on score). The total number of MS/MS fragmentation spectra (Spectral count or SpC) was used for subsequent protein quantification in the different samples. All data have been deposited to the ProteomeXchange Consortium [88].

### Exploratory and differential expression analysis of LC-MS/MS data

Mass spectrometry data obtained for each sample were stored in a local MongoDB database and subsequently analyzed through a Shiny Application built upon the R/Bioconductor packages msmsEDA (Gregori J, Sanchez A, Villanueva J (2014). msmsEDA: Exploratory Data Analysis of LC-MS/MS data by spectral counts. R/Bioconductor package version 1.22.0) and msmsTests (Gregori J, Sanchez A, Villanueva J (2013). msmsTests: LC-MS/MS Differential Expression Tests. R/Bioconductor package version 1.22.0). Exploratory data analyses of LC-MS/MS data were thus conducted, and differential expression tests were performed using a negative binomial regression model. The p-values were adjusted with FDR control by the Benjamini-Hochberg method and the following criteria were used to define differentially expressed proteins: an adjusted p-value < 0.05, a minimum of 5 SpC in the most abundant condition, and a minimum fold change of 2 (abs(LogFC) > 1). GO term analysis was performed using the EnrichR web-based tool (http://amp.pharm.mssm.edu/Enrichr). The direct interaction network for proteins enriched in SINV-infected cells was generated using the STRING database (https://string-db.org).

### Supporting information

**S1 Fig. Analysis of SINV-GFP infection in FHA:DICER cell lines at different MOI and time points. A.** miR-16 expression analyzed by northern blot in HEK293T, NoDice FHA:ctrl #1 and FHA:DICER WT #4 cell lines. Expression of snRNA U6 was used as loading control. **B.**

Representative GFP pictures of HEK293T, NoDice 2.20, NoDice 4.25, NoDice FHA:ctrl #1 and NoDice FHA:ctrl #2 cells infected with SINV-GFP at an MOI of 0.02 for 24 h. The left panel corresponds to GFP signal and the right panel to a merge of GFP signal and the corresponding brightfield. Pictures were taken with a 5x magnification. hpi: hours post-infection. **C.** Mean (+/- SEM) of SINV-GFP viral titers in cells infected at an MOI of 0.02 for 24 h (n = 3) from plaque assay quantification. $^*$ p < 0.05, ns: non-significant, ordinary one-way ANOVA test with Bonferroni correction. **D**. Western blot analysis of DICER, AGO2 and GFP expression in SINV-GFP-infected cells shown in B. Gamma-Tubulin was used as loading control. (TIF)

**S2 Fig. LC-MS/MS analysis of DICER interactome during SFV infection. A.** Volcano plot for differentially expressed proteins (DEPs) between HA IP and CTL IP in FHA:DICER mock-infected cells. Each protein is marked as a dot; proteins that are significantly up-regulated in HA IP are shown in red, up-regulated proteins in CTL IP are shown in blue, and non-significant proteins are in black. The horizontal line denotes a p-value of 0.05 and the vertical lines the Log2 fold change cutoff (-1 and 1). DICER and its cofactors (TRBP, PACT, AGO2) are highlighted in yellow. **B.** Left panel: Volcano plot for DEPs between SFV (MOI of 2, 6 hpi) and mock fractions of HA IP in FHA:DICER cells. Same colour code and thresholds as in A were applied. Proteins that are discussed in the text are highlighted in yellow and SFV proteins in purple. **C.** Summary of the differential expression analysis of SFV vs mock fractions from HA IP in FHA:DICER cells. The analysis has been performed using a generalized linear model of a negative-binomial distribution and p-values were corrected for multiple testing using the Benjamini-Hochberg method. (TIF)

**S3 Fig. Confirmation of LC-MS/MS analysis by co-IP and BiFC controls. A.** FHA:DICER WT #4 cells were infected with SINV-GFP at an MOI of 0.02 for 24 h and a HA co-IP was performed. Eluted proteins were resolved by western blot and IP efficiency was assessed using an HA antibody. In parallel, co-IPed proteins were visualized using appropriate antibodies. GFP antibody was used to verify the infection and Ponceau staining serves as loading control. **B.** 1% agarose gel analysis of RNA extracted from INPUT of the co-IP in Fig 3B. Ribosomal RNA integrity was compared to a control HEK293T cell line. RNAs were revealed using ethidium bromide under UV. **C.** Schematic representation of Human DICER proteins used for BiFC positive and negative controls. The different conserved domains are shown in colored boxes. DUF283: Domain of Unknown Function; PAZ: PIWI ARGONAUTE ZWILLE domain; dsRBD: dsRNA-binding domain. hDICER WT is the full-length protein. hDICER N1 is deleted of the first N-terminal 495 amino acids. **D.** Expression of BiFC plasmids was assessed by western blot. DICER proteins (WT and N1) and PKR were visualized using antibodies targeting endogenous proteins, whereas TRBP and PACT were detected using GFP antibody. Antibody targeting the SINV coat protein (CP) was used as infection control. Ponceau staining was used as loading control. **E.** Positive and negative BiFC controls on fixed NoDiceΔPKR cells. After co-transfection, cells were infected with SINV at an MOI of 2 for 6 h and fixed. After fixation, cells were stained with DAPI and observed under confocal microscope. Merge pictures of BiFC and DAPI signals of SINV-infected cells are shown. A higher magnification of images showing the interaction represented by a red square is shown in the bottom left corner. Scale bars: 20 μm and 10 μm. **F.** Expression of BiFC plasmids was assessed by western blot. DICER, PKR, TRBP and PACT were detected using GFP antibody. Antibody targeting the SINV coat protein (CP) was used as infection control. Gamma-Tubulin was used as loading control. The asterisk corresponds to an aspecific band. **G.** Interactions between DICER and TRBP, PACT or PKR were visualized by BiFC. Plasmids expressing $^{N\text{-}ter}$Venus:DICER

and TRBP:, PACT: or PKR:Venus$^{C-ter}$ were co-transfected in HEK293T cells for 24 h and cells were either infected with SINV at an MOI of 2 for 6 h or not. The different combinations are indicated on the left side. Reconstitution of Venus (BiFC) signal was observed under epifluorescence microscope. For each condition, the left panel corresponds to Venus signal and the right panel to the corresponding brightfield pictures. Scale bar: 100 μm.
(TIF)

**S4 Fig. Confirmation of DICER interactome upon SINV infection in HCT116 KI-DICER cells. A.** Schematic representation of DICER WT and Flag-HA(FHA)-GFP knocked-in (KI) alleles. FHA sequence is in purple, GFP in green, DICER 5'UTR in orange and DICER coding region in yellow. The gRNA used to generate the KI was designed to target the first coding exon of DICER gene. **B.** PCR on genomic DNA extracted from WT and KI cells. **C.** An oligo outside the homologous recombination region and an oligo within the GFP tag were used to verify the presence of a 1040 bp amplicon in HCT116 KI-DICER clone. Sequencing results corresponding to this region are shown. **D.** Western blot analysis of DICER, p-PKR, PKR and p-eIF2α expression in mock or SINVGFP-infected HEK293T and HCT116 KI-DICER cell lines at an MOI of 2 for 6 h or 16 h and 0.02 for 24 h. GFP antibody was used to verify the infection. Ponceau and gamma-Tubulin were used as loading controls.
(TIF)

**S5 Fig. Interaction analysis between the different versions of DICER and TRBP or PACT using BiFC assay.** NoDiceΔPKR cells were co-transfected for 24 h with plasmids expressing the different versions of DICER proteins fused to the N-terminal part of Venus and either TRBP:Venus$^{C-ter}$ (**A**) or PACT:Venus$^{C-ter}$ (**B**). Cells were then infected with SINV at an MOI of 2 for 6 h and Venus signal was observed under epifluorescence microscope. The left panel corresponds to Venus signal and the right panel to the corresponding brightfield picture. Pictures were taken with a 5x magnification. hpi: hours post-infection. Scale bar: 100 μm.
(TIF)

**S6 Fig. Analysis of the importance of Dicer helicase domain on SINV-GFP infection in FHA:DICER mutant stable cell lines. A**. Northern blot analysis of miR-16 expression in HEK293T, NoDice 2.20, NoDice FHA:ctrl #2, FHA:DICER WT polyclonal, FHA:DICER N1 #6, FHA:DICER Hel. #2.6, and FHA:DICER N3 #2.13. Expression of snRNA U6 was used as loading control. **B.** Representative GFP fluorescent microscopy images of HEK293T, NoDice 2.20, FHA:DICER mutants cell lines infected with SINV-GFP at an MOI of 0.02 for 24 h. The left panel corresponds to GFP signal and the right panel to a merge picture of GFP signal and brightfield. Pictures were taken with a 5x magnification. hpi: hours post-infection. **C.** Mean (+/- SEM) of SINV-GFP viral titers over FHA:DICER WT #4 cells in FHA:DICER N1 #6, FHA:DICER N3 #2.13, NoDice FHA:ctrl #2 and NoDice 2.20 cell lines infected at an MOI of 0.02 for 24 h (n = 3) from plaque assay quantification. $^{***}$ p < 0.001, ns: non-significant, ordinary one-way ANOVA test with Bonferroni correction.
(TIF)

**S1 Table. Top 100 proteins that are differentially immunoprecipitated in mock-infected FHA:DICER cells by the HA and Myc (CTL) antibodies.** Related to Fig 2.
(XLSX)

**S2 Table. Top 100 proteins that are differentially immunoprecipitated with the HA antibody in SINV-infected vs mock-infected FHA:DICER cells.** Related to Fig 2.
(XLSX)

**S3 Table. Top 100 proteins that are differentially immunoprecipitated in mock-infected FHA:DICER cells by the HA and Myc (CTL) antibodies, in the SFV infection experiment.** Related to S2 Fig.
(XLSX)

**S4 Table. Top 100 proteins that are differentially immunoprecipitated with the HA antibody in SFV-infected vs mock-infected FHA:DICER cells.** Related to S2 Fig.
(XLSX)

**S5 Table. List of primers used in this study.**
(XLSX)

**S6 Table. Data and statistical tests details used in plaque assays shown in Figs 1, 6, 7, 8, S1 and S6.**
(XLSX)

## Acknowledgments

The authors would like to thank members of the Pfeffer laboratory for discussion, Pr. Bryan Cullen for the kind gift of the HEK293T NoDice and NoDiceΔPKR cell lines and Dr. Oliver Vugrek for the BiFC plasmids.

## Author Contributions

**Conceptualization:** Thomas C. Montavon, Mathieu Lefèvre, Sébastien Pfeffer.

**Data curation:** Thomas C. Montavon, Philippe Hammann, Johana Chicher.

**Formal analysis:** Thomas C. Montavon, Morgane Baldaccini, Mathieu Lefèvre, Erika Girardi, Philippe Hammann, Johana Chicher, Sébastien Pfeffer.

**Funding acquisition:** Sébastien Pfeffer.

**Investigation:** Morgane Baldaccini, Mathieu Lefèvre, Erika Girardi, Mélanie Messmer.

**Methodology:** Sébastien Pfeffer.

**Project administration:** Sébastien Pfeffer.

**Software:** Béatrice Chane-Woon-Ming.

**Supervision:** Sébastien Pfeffer.

**Validation:** Thomas C. Montavon, Morgane Baldaccini, Erika Girardi, Mélanie Messmer.

**Visualization:** Béatrice Chane-Woon-Ming.

**Writing – original draft:** Thomas C. Montavon, Morgane Baldaccini, Sébastien Pfeffer.

**Writing – review & editing:** Thomas C. Montavon, Morgane Baldaccini, Mathieu Lefèvre, Erika Girardi, Béatrice Chane-Woon-Ming, Mélanie Messmer, Sébastien Pfeffer.

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
