## [Decision Letter · Decision Letter 0]

25 Jan 2021

Dear Dr. Pfeffer,

Thank you very much for submitting your manuscript "Human DICER helicase domain recruits PKR and dsRNA binding proteins during viral infection" for consideration at PLOS Pathogens. As with all papers reviewed by the journal, your manuscript was reviewed by members of the editorial board and by several independent reviewers. In light of the reviews (below this email), we would like to invite the resubmission of a significantly-revised version that takes into account the reviewers' comments.

Please carefully address all of the comments provided by the reviewers. Experimentally, I would suggest focusing on the major comments of reviewer #2 to address the mechanisms underlying the antiviral activity of the Dicer helicase mutants.

We cannot make any decision about publication until we have seen the revised manuscript and your response to the reviewers' comments. Your revised manuscript is also likely to be sent to reviewers for further evaluation.

Sincerely,

Stacy M Horner

Associate Editor

PLOS Pathogens

Mark Heise

Section Editor

PLOS Pathogens

Kasturi Haldar

Editor-in-Chief

PLOS Pathogens

orcid.org/0000-0001-5065-158X

Michael Malim

Editor-in-Chief

PLOS Pathogens

orcid.org/0000-0002-7699-2064

Reviewer's Responses to Questions

**Part I - Summary**

Reviewer #1: This is a modified version of the eLife manuscript that I previously reviewed. This new version tones down conclusions and adds further data, crucially supporting their original observations in different mammalian cell liens and impressively under endogenous expression conditions. This current manuscript provides a useful description of a previously unknown connection between the RNAi and innate antiviral defenses in mammals.

Reviewer #2: This manuscript aims to investigate and clarify an antiviral function of human Dicer. For this aim, the authors performed proteomics analysis and demonstrated the Dicer’s interactome during Sindbis virus (SINV) infection. They successfully identified 184 proteins as proteins that specifically interact with Dicer during SINV infection. Among them, they focused on the interaction with dsRNA-binding proteins and RNA helicases such as TRBP, PKR, PACT, ADAR1, and DHX9. The main aim of this study is very interesting, but I am concerned that several important analyses/discussions are missing.

Reviewer #3: This study aims to gain insight into the role of the ribonuclease Dicer on viral infection of human cells. The authors used a proteomic approach to identify the cellular proteins that are interacting with Dicer upon infection with two RNA viruses: Sindbis virus (SINV) and Semliki Forest virus (SFV). They first complemented HEK293T Dicer knock-out cell lines (NoDice cells) with a lentiviral vector-expressing Flag-HA tagged human Dicer (FHA:Dicer) and pursued their study using a clone in which the miRNA biogenesis was restored and the permissivity to SINV was comparable to the parental HEK 293T cells. They then performed an HA-immunoprecipitation (IP) followed by a mass spectrometry analysis to identify the Dicer interactome upon viral infection. Among the Dicer-interacting proteins enriched in infected cells, they identified many double-stranded RNA (dsRNA) binding proteins known to play a role in the interferon pathway such as PKR, ADAR1, DHX9 and PACT. They focused their study on the characterisation of Dicer-PKR interaction and found that it was highly, albeit not fully, RNA-dependent and confirmed that this interaction was occurring in vivo using bi-molecular fluorescent complementation assay (BiFC). They further show that the Dicer helicase domain is required for its interaction with PKR. While the helicase domain was necessary and sufficient to bind the non-phosphorylated form of PKR, this was not the case for the interaction with the phosphorylated form of PKR, which additionally depends on the dsRNA-binding domain of Dicer. Finally, the authors show that the expression of truncated forms of Dicer lacking part or all of the helicase domain confer a strong antiviral activity against SIV-GFP infection.

This study is well written, the experiments are well-conducted and the conclusions are supported by the data. The findings are interesting and important for the field as it provides further understanding of the role of Dicer during a viral infection and its intriguing interplay with components of the interferon pathway. However, I have few comments that should be addressed.

**Part II – Major Issues: Key Experiments Required for Acceptance**

Reviewer #1: (No Response)

Reviewer #2: 1. The authors infected GFP-fusion SINV into HEK293T cells or Dicer knock-out cells and detected the fluorescence for the detection of viral replication. The human genome encodes single Dicer, so I am wondering whether the knock-out of Dicer influences the speed of cell growth. Does the difference of cell growth between WT and Dicer knock-out cells influence the viral replication? Is the decrease of fluorescence observed in Dicer knock-out cells (Fig.1A) a direct effect by Dicer for the regulation of viral replication?

2. The major effort of this manuscript seems to perform and describe clearly the results of immunoprecipitation and the BiFC approach. However, the most intriguing result with a lot of information is the proteomic analysis on Fig.2. The authors focused on PKR, PACT, ADAR1, and DHX9 in the subsequent analysis, but is it possible to look into the proteomic results more, not only these proteins but also others. My suggestion does not mean the requirement of additional experiments but might need additional informatical analysis such as GO analysis or Venn plot.

3. Through the manuscript, there are several contradictions. For example, Dicer-PACT/Dicer-PKR interaction (Fig.3A vs Fig.3D), Dicer-TRBP interaction (Fig.3A vs Fig.4D), and GFP fluorescence level of FHA:ctrl #1 and FHA:Dicer WT#4 (Fig.1A vs Fig.6E) are inconsistent.

4. In line 366, the authors mentioned that the deletion of dsRNA-binding domain did not influence the interaction of Dicer-PACT or Dicer-PKR, but RNase treatment decreased their interaction in Fig. 3B. Is dsRNA necessary for their interaction, or not?

5. The immunoprecipitation of endogenous PKR and Dicer/PACT (Fig.3C) might be changed to a clearer result.

Reviewer #3: While the interaction between Dicer and PKR and its mapping to Dicer helicase domain is compelling, this study doesn’t provide mechanistic insight into the role of Dicer-PKR interaction during viral infection. The authors demonstrated that the Dicer helicase domain was required for the interaction with PKR (Fig.5B,C and D). They then nicely showed that expression of Dicer helicase deletion mutants (Dicer N1 and Dicer N3) in NoDice HEK293T confer antiviral activity against SINV-GFP (Fig.6C,D and E). However, there is no experiments addressing the mechanism by which Dicer N1 and Dicer N3 exert their antiviral activity and importantly whether it is dependent or not on PKR. This reviewer appreciates that a full dissection of the mechanisms at play is beyond the scope of this study. However, two experiments seems feasible to further investigate the mechanisms by which Dicer N1/N3 gained its antiviral activity.

• One possibility is that Dicer binds PKR through its helicase domain and thereby inhibits PKR antiviral activity. The expression of Dicer N1/N3, which are unable to bind PKR, would lead to an increased activation of PKR during viral infection. The authors should therefore test whether the expression of Dicer N1/N3 exhibits similar antiviral activities in the absence of PKR. This could be done by stably expressing these Dicer helicase deletion mutants in NoDice/PKR KO cells (used in all BiFC experiments) and test whether or not they still exhibit antiviral activity against SINV-GFP.

• Previous studies showed that the deletion of the helicase domain provides Dicer with an increased ability to process long dsRNA into siRNA and that expression of Dicer N1/N3 results in the accumulation of viral siRNA upon infection with Influenza virus (Kennedy et al., 2016; Tsai et al., 2018). The authors should therefore test whether the observed antiviral effect of Dicer N1/N3 on SINV-GFP is dependent on the RNAi pathway. An informative experiment would be to knock-down (or knock-out) Ago2 in NoDice FHA:Dicer N1 # 6 and NoDice FHA:Dicer N3 #2.13 (Fig. 6D and E) and test whether the antiviral activity of Dicer N1/N3 against SINV-GFP is impacted or not by the inactivation of the RNAi pathway.

**Part III – Minor Issues: Editorial and Data Presentation Modifications**

Reviewer #1: This new version still sets up well the ongoing questions of whether RNAi is a relevant antiviral response in mammals, but exactly how the current work fits into this overall field was obscure. I believe the authors simply want to describe a novel interaction between RNAi and canonical mammalian antiviral response, and that Dicer could even be inhibitory in some situations. But, this could be made more clear in the Abstract/Discussion, and if I am missing some other points regarding this central question, perhaps these could also be stated directly in a more straightforward manner.

Reviewer #2: (No Response)

Reviewer #3: • The authors nicely show that the interactions between Dicer and PKR, p-PKR, DHX9 and PACT are sensitive to RNAse A/T1 treatment demonstrating that these interactions are RNA-dependent. RNAse A/T1 specifically digest single-stranded RNAs, while PKR, ADAR1, TRBP, PACT are dsRNA-binding proteins. It would benefit the study to include an experiment in which the extracts are treated with a dsRNA-specific RNAse (RNase III) to test whether these interactions that are induced upon viral infection also depend on the accumulation of dsRNA generated during viral replication.

• Page 7, line 146: the authors mention that “Dicer antiviral effect is not reproducible in an independent clone”, but the data shows a Dicer proviral effect in some of the clones rather than an antiviral activity. The GFP signal from SINV-GFP infection is indeed reduced (and not increased) upon infection of NoDice cells 2.20 transduced or not with a lentiviral control.

• Fig.S3D: The authors should show a larger portion of the immunoblot probed with anti-GFP antibody in order to also show the level of expression of PKR:Venus Cter.

• Fig.S4D: the Ponceau staining is not informative in this case as it doesn’t show much proteins in each well. As a loading control using anti-tubulin antibody was also performed and included in the figure, the authors should consider removing the Ponceau staining from this figure.

• Figure 4D: the immunoblot showing the co-IP of Dicer with TRBP is not placed together with the co-IP of Dicer with PKR, PACT and DHX9 suggesting that these are 2 independent experiments. If it is the case, the level of Dicer (probed by an anti-HA antibody) in all fractions should also be included for this independent experiment. If these two sets of immunoblots are not independent experiments but are two membranes from the same experiment, this should be specified in the figure legend.

• Figure 5B: there is a band visible in all wells of the “HA-IP” probed with anti-GFP antibody while all samples are not infected with SINV-GFP.

• Figure 5C: the molecular weight marker on the left-hand side of the immunoblot showing the input samples probed with anti-FLAG antibody might be incorrect as it shows that Dicer N3 runs at 70kDa. The “70kDa” label is likely part of the immunoblot just below and there is another “250 kDa” label under the “70kDa” that is likely a mistake and should be removed. The immunoblot of input samples probed with anti-FLAG antibody and showing the proteins in the range of 70kDa is not clear as it is difficult to see the expression of the “Dicer Hel.” construct among the additional bands.

• Typo: Page 3, line 53: interferon-stimulated genes.

PLOS authors have the option to publish the peer review history of their article (what does this mean?). If published, this will include your full peer review and any attached files.

Reviewer #1: No

Reviewer #2: No

Reviewer #3: No
---

## [Editor Report · Decision Letter 1]

8 Apr 2021

Dear Dr. Pfeffer,

We are pleased to inform you that your manuscript 'Human DICER helicase domain recruits PKR and modulates its antiviral activity' has been provisionally accepted for publication in PLOS Pathogens.

Best regards,

Stacy M Horner

Associate Editor

PLOS Pathogens

Mark Heise

Section Editor

PLOS Pathogens

Kasturi Haldar

Editor-in-Chief

PLOS Pathogens

orcid.org/0000-0001-5065-158X

Michael Malim

Editor-in-Chief

PLOS Pathogens

orcid.org/0000-0002-7699-2064
---

## [Editor Report · Acceptance letter]

16 Apr 2021

Dear Dr. Pfeffer,

We are delighted to inform you that your manuscript, "Human DICER helicase domain recruits PKR and modulates its antiviral activity," has been formally accepted for publication in PLOS Pathogens.

Best regards,

Kasturi Haldar

Editor-in-Chief

PLOS Pathogens

orcid.org/0000-0001-5065-158X

Michael Malim

Editor-in-Chief

PLOS Pathogens

orcid.org/0000-0002-7699-2064